# Geological and Petrophysical Properties of Underground Gas Storage Facilities in Ukraine and Their Potential for Hydrogen and CO₂ Storage

Yuliia Demchuk [1,*], Kazbulat Shogenov [2,3,*], Alla Shogenova [2,3], Barbara Merson [4] and Ceri Jayne Vincent [5]

[1] NGO "Geothermal Ukraine", 76000 Ivano-Frankivsk, Ukraine
[2] SHOGenergy Consulting, 11414 Tallinn, Estonia; alla.shogenova@taltech.ee
[3] Department of Geology, Tallinn University of Technology, 19086 Tallinn, Estonia
[4] National Institute of Oceanography and Applied Geophysics—OGS, 34010 Sgonico, TS, Italy; bmerson@ogs.it
[5] British Geological Survey, Keyworth, Nottingham NG12 5GG, UK; cvi@bgs.ac.uk
* Correspondence: y.demchuk@geothermalukraine.org (Y.D.); kazbulat.shogenov@taltech.ee (K.S.)

**Abstract:** This article provides detailed geological and reservoir data on the existing underground gas storage (UGS) facilities in Ukraine and their prospects for hydrogen (H₂) and carbon dioxide (CO₂) storage. The H₂ and CO₂ storage issue is an integral part of the decarbonisation of Ukraine and Europe as a whole. A detailed assessment of UGS in Ukraine was carried out in the framework of the EU Horizon 2020 project Hystories, which is about the possibility of the geological storage of H₂. A database of the available geological data on reservoir and caprock properties was compiled and standardised (reservoir geometry, petrophysics, tectonics, and reservoir fluids). General environmental criteria were defined in terms of geology and surface context. The total estimated H₂ energy storage capacity in 13 studied UGS facilities is about 89.8 TWh, with 459.6 and 228.2 Mt of H₂ using the total (cushion and working gas) and working gas volumes, respectively. The estimated optimistic and conservative CO₂ storage capacities in the 13 studied UGS facilities are about 37.6/18.8 Gt, respectively. The largest and deepest UGS facilities are favourable for H₂ and CO₂ storage, while shallower UGS facilities are suitable only for H₂ storage. Studies could be conducted to determine if CO₂ and H₂ storage could be applied in synergy with CO₂ being used as a cushion gas for H₂ storage. The underground storage of H₂ and CO₂ plays key roles in reducing greenhouse gas emissions and supporting clean energy while enhancing energy security. Increasing the share of renewable energy and integrating sustainable development across various sectors of the economy is crucial for achieving climate goals.

**Keywords:** underground gas storage; hydrogen storage; CO₂ geological storage; cushion gas; reservoir rocks; sustainable decarbonisation

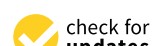

## 1. Introduction

Ukraine, as an active member of the UN Framework Convention on Climate Change (UNFCCC), was among the first nations on the European continent to ratify the Paris Agreement on 14 July 2016 and the Kyoto Protocol on 4 February 2004. Ukraine, as a candidate for accession to the European Union (EU), aims to align with EU climate goals, including the European Climate Law's target of climate neutrality by 2050 and a 55% greenhouse gas emissions reduction by 2030. As a country that ratified an Association Agreement with the EU on 16 September 2014 and that is a member of the Energy Community Treaty, Ukraine is dedicated to

(1)   Reduce the total amount of greenhouse gas emissions (GHGEs) by 65% by 2030, compared to 1990 levels;

(2)   Increase the share of renewable energy sources in the structure of gross final energy consumption to not less than 27% by 2030;

(3)   Become climate neutral by 2060, in line with the long-term ambitions of EU members and in accordance with EU policies [1].

In response to the hardships and global energy market disruption caused by Russia's invasion of Ukraine, the EU is implementing its REPowerEU Plan [2] in February 2023, focusing on renewable energy and thereby reducing reliance on Russian gas and the effects of climate change. The energy sector is responsible for more than 75% of the EU's GHGEs. The revised Renewable Energy Directive EU/2023/2413 raises the EU's binding renewable target for 2030 to a minimum of 42.5% of consumption, up from the previous 32% target, with the aspiration to reach 45% [3]. Decarbonisation via carbon dioxide ($CO_2$) geological storage and hydrogen ($H_2$) production, part of the decarbonisation pillar of the EU Green Deal and Fit-for-55 package REPowerEU plan, will reduce $CO_2$ emissions, enhance energy security, and create new jobs.

To achieve these goals, Ukraine aims to apply innovative technologies to decarbonise its energy sector and boost energy efficiency. Ukraine's integration into the EU's renewable and decarbonisation frameworks can enhance cooperation and contribute to achieving climate neutrality and sustainable development.

The challenges in decarbonising the energy sector are reported with respect to environmental sustainability, the security of the energy supply, economic stability, and social aspects in [4]. A global carbon tax was reported as the most promising, but very challenging, instrument for humanity to accelerate the process of decarbonisation.

Potential solutions include $H_2$ energy and $CO_2$ capture, utilisation, and storage (CCUS) technologies. To ensure energy security and reduce $CO_2$ emissions, Ukraine's existing underground gas storage (UGS) facilities, currently used for surplus European gas storage, could be repurposed for $H_2$ and $CO_2$ storage, efficiently using existing infrastructure. It is, therefore, important to conduct a comprehensive assessment of Ukraine's existing UGS facilities to estimate their capacity for $H_2$ and $CO_2$ storage, as no comprehensive studies have been conducted in Ukraine.

UGS facilities are a special case as they could potentially be converted directly into $H_2$ storage sites since they are already connected to the gas network and have already been well characterised, and pressure and fluid movement in the reservoir is well understood. Although numerous research projects are investigating the potential of underground $H_2$ storage (UHS), the number of fully operational projects in UGS facilities remains limited. There are existing UHS facilities in salt caverns in Teesside (UK), Spindletop (USA), and Clemens Dome and Moss Bluff (USA). Hydrogen-rich gas was stored in an aquifer in Lobodice (Czech Republic) and in a UGS store in a depleted gas reservoir in Austria. Pure $H_2$ storage in a depleted gas reservoir started during 2023 in Austria (the Underground Sun Storage project) [5]. Global experience indicates that $H_2$ storage poses significant safety risks due to its unique properties, such as its high flammability, low molecular weight, and the potential for leaks. As a result, robust safety measures and modelling are essential to mitigate these risks [6]. Extensive experience from many countries on UGS offers important relevant experience in managing these risks. Building on experience from projects like Sleipner [7] and Snøhvit [8] (Norway), UGS facilities also offer a significant potential for long-term $CO_2$ sequestration. There is currently no experience of $CO_2$ storage or UHS in Ukraine.

Assuming the cushion gas, which comprises natural gas, is retained during $H_2$ storage and neglecting any mixing issues, UGS stores offer a promising opportunity for $H_2$ storage.

Alternatively, $CO_2$ and $H_2$ storage could be used synergistically if $CO_2$ is used as a cushion gas for $H_2$ storage (Figure 1). More detailed, site-specific simulations of this potential mixing are needed to determine the feasibility of using $CO_2$ as a cushion gas.

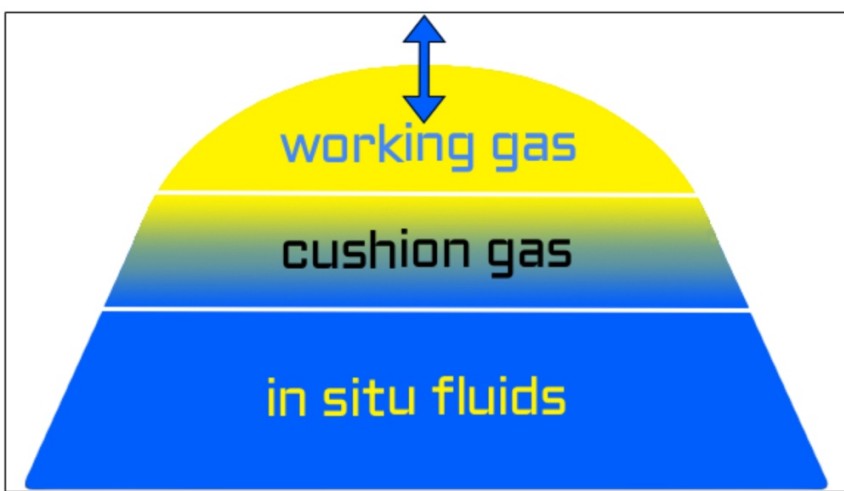

**Figure 1.** Diagram showing gas distribution within a geological storage site (modified from [9]). The bidirectional nature of the arrow signifies that the working gas is injected and withdrawn as needed.

Accurately estimating the $CO_2$ storage capacity allows potential emissions reductions to be quantified and sensible storage targets to be made. This information is crucial for demonstrating the feasibility and effectiveness of CCUS projects in achieving emission reduction targets.

The objective of this article is to analyse the geological and petrophysical properties of UGS facilities, estimate their potential for $H_2$ and $CO_2$ storage, and assess possible synergies. The careful estimation of $H_2$ and $CO_2$ storage capacity is a complex process that requires a multidisciplinary approach and data analyses (reservoir and caprock lithological characterisation, a porosity and permeability assessment, and the evaluation of pressure and temperature conditions) [10].

As a part of the H2020 EU project "HYdrogen STORage in European Subsurface" (Hystories [11]), aimed at supporting technical developments for the storage of pure green $H_2$ in depleted fields and aquifers, a thorough analysis was conducted on 13 UGS facilities in Ukraine. The purpose of the Hystories project was to assess the potential of these sites for the geological storage of $H_2$. During the Hystories project, geochemical reactions and microbiological impacts and mitigations were studied [12,13]. Given the potential for geochemical and microbiological reactions concerning UHS, site-specific studies would be required to confirm suitability.

Ukrainian UGS facilities were also included in a recently published techno-economic modelling study of $H_2$ storage in European UGS facilities [14], in which only working gas volume data of UGS facilities, reported in 2004, were collected and used for calculations, and the total volumes of all the European UGS facilities were not available.

The study presented here used and analysed the total, working, and cushion gas volumes in Ukrainian UGS facilities for $H_2$ and $CO_2$ storage and their possible synergy.

## 2. Geological Background and Location of UGS in Ukraine

The Ukrainian mainland comprises three major oil- and gas-bearing basins in the western, eastern, and southern regions (Figure 2). Existing UGS facilities, as well as most depleted oil and gas fields, belong to these basins and can be considered potentially suitable for $H_2$ and/or $CO_2$ storage.

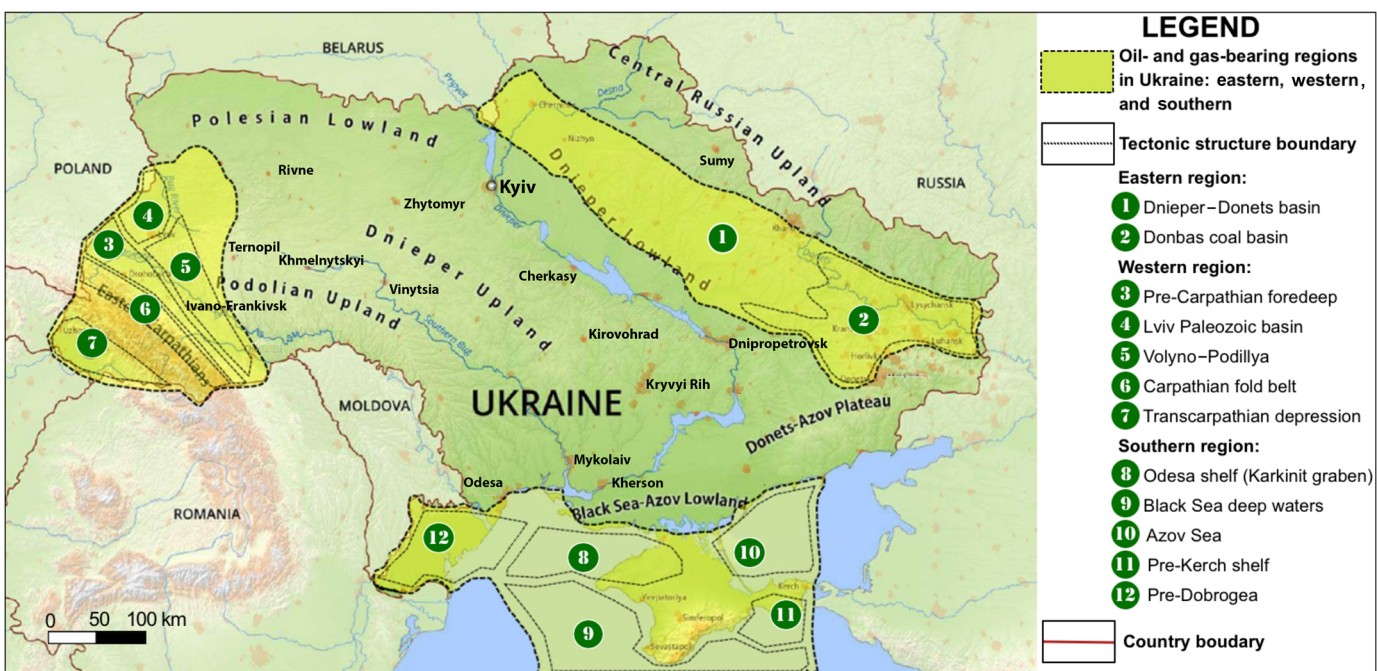

**Figure 2.** Oil- and gas-bearing regions in Ukraine. Modified from [15].

Ukraine has a developed network of 13 UGS facilities with a total capacity of more than 31 billion cubic metres (BCMs) of natural gas. Eleven of the UGS facilities were built in depleted gas and gas condensate fields (Uherske, Bilche–Volytsko–Uherske, Oparske, Dashavske, Bohorodchanske, Solokhivske, Kehychivske, Proletarske, Krasnopopivske, Verhunske, and Hlibivske), and two were built in aquifers (Olyshivske and Chervonopartyzanske) (Figure 3 [16]).

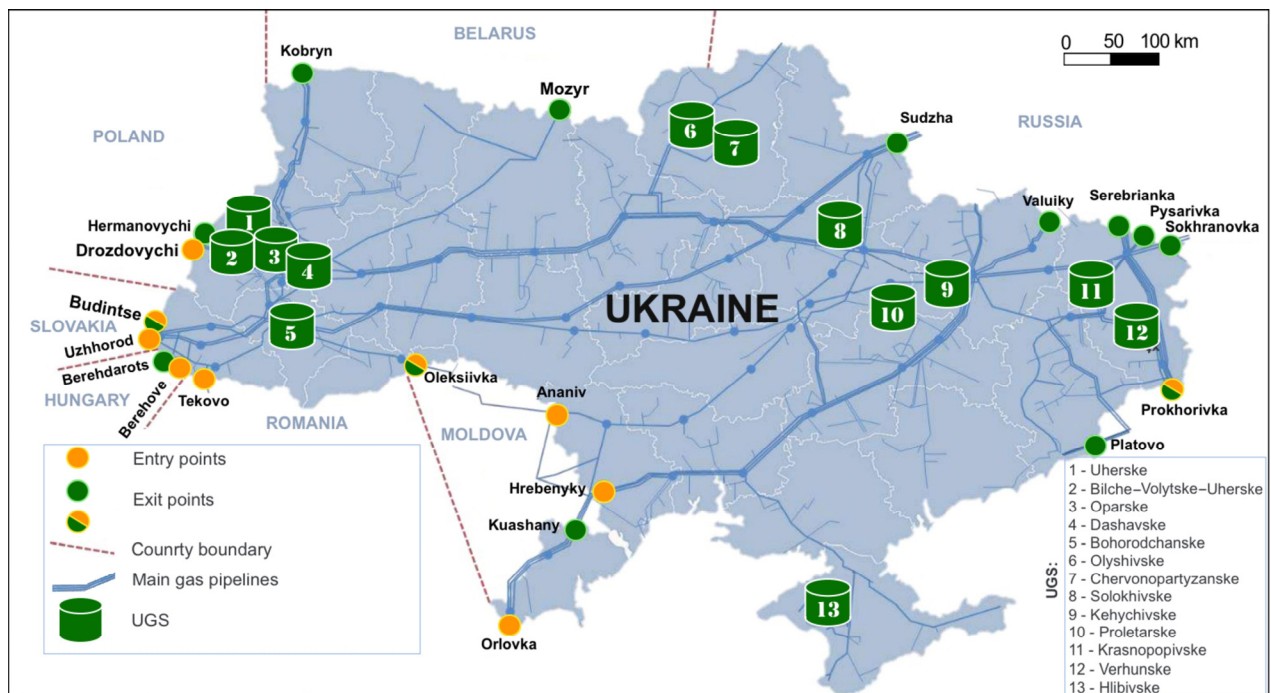

**Figure 3.** Location of existing UGS facilities in Ukraine. Modified from [16].

The significant capacities of the existing onshore UGS facilities lie at depths ranging from 400 to 2000 m, making it possible to consider the UGS system as an early opportunity in Ukraine for $H_2$ storage or for $CO_2$ storage in deeper reservoirs. There are four UGS complexes

which can be defined by the location of the UGS facility and its connection to the main gas pipelines in Ukraine: (1) western, (2) central, (3) eastern, and (4) southern (Figure 4).

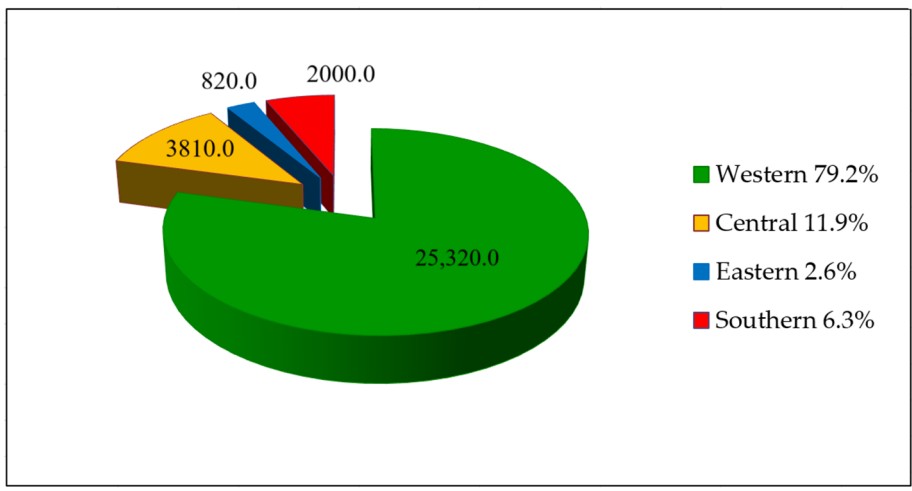

**Figure 4.** The distribution of the working gas volume across the UGS complexes, Mm$^3$.

### 2.1. Western Region

The western UGS region is located in the Carpathian Foreland (Figures 5–7) and is connected to a transcontinental, interstate, and intrastate gas pipeline system. There are five gas storage facilities:

1. Uherske.
2. Bilche–Volytske–Uherske.
3. Oparske.
4. Dashavske.
5. Bohorodchanske.

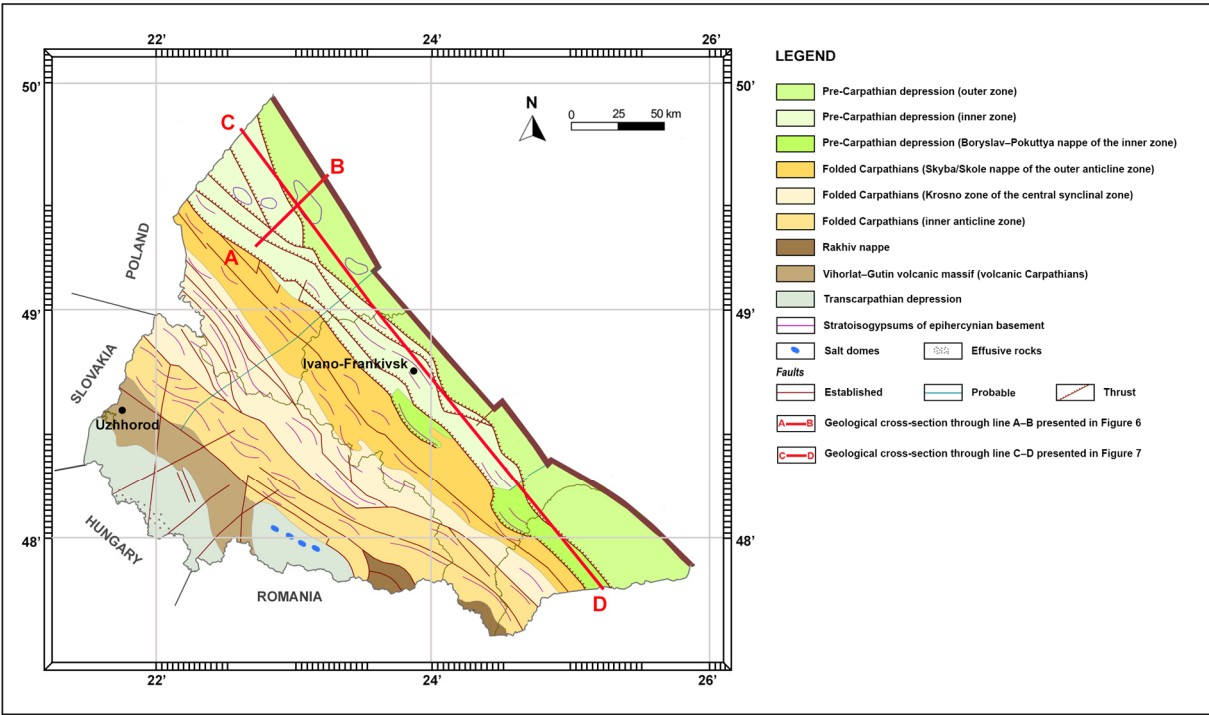

**Figure 5.** Tectonic map of the Ukrainian Carpathians (modified from [17]).

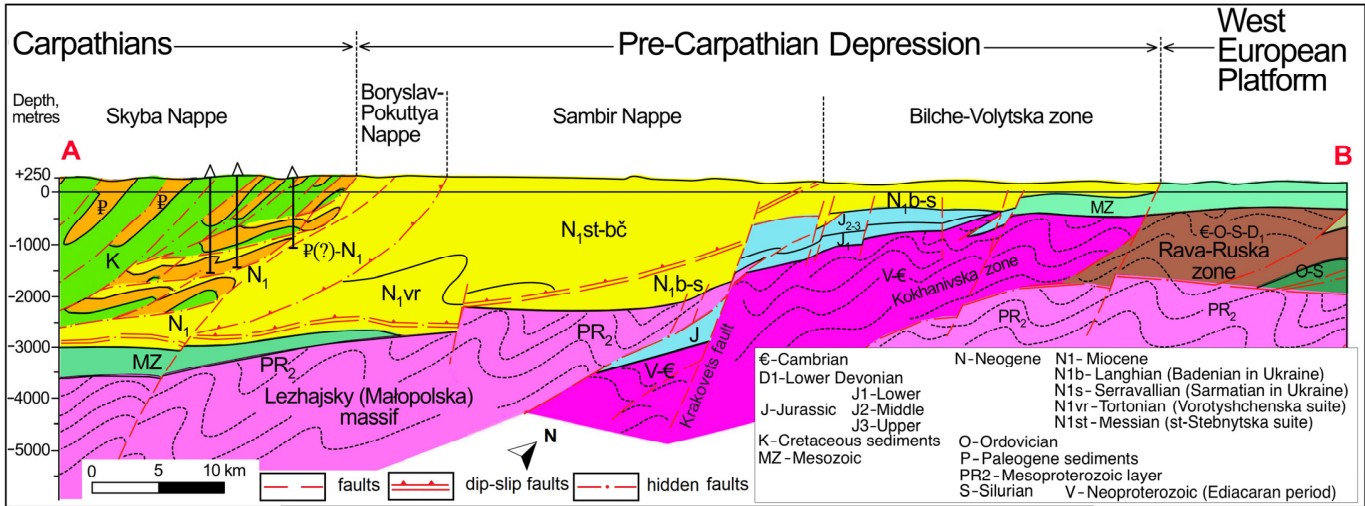

**Figure 6.** Geological cross-section through the A–B line indicated in Figure 5 (modified from [18]).

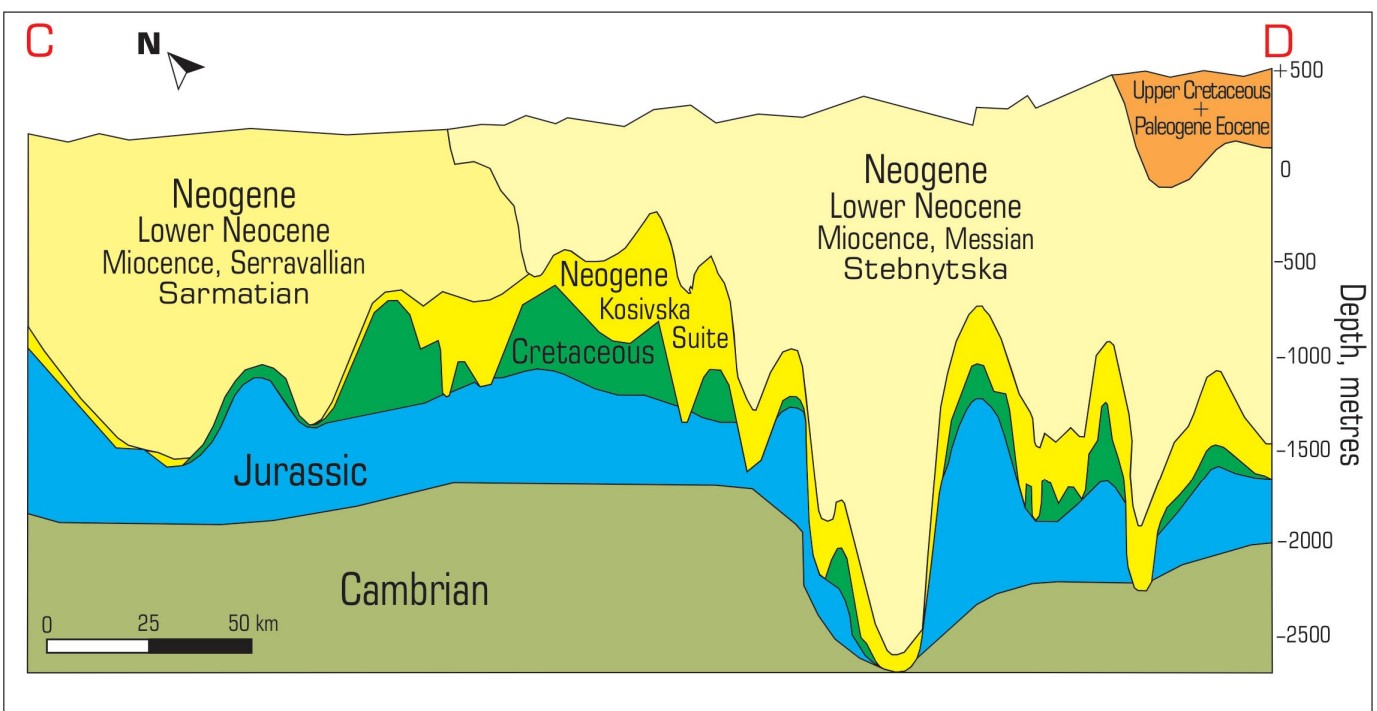

**Figure 7.** Geological cross-section through line C–D, line indicated in Figure 5 (modified from [19]).

The UGS facilities of western Ukraine are limited to the Bilche–Volytska zone—the outer part of the Pre-Carpathian Depression. The structural–tectonic Bilche–Volytska zone is one of western Ukraine's most important gas-producing zones. It was formed on the submerged southwestern edge of the Eastern European Platform, where layers and lenses of sandstones and siltstones formed traps for hydrocarbon deposits under different geodynamic conditions [20]. Massive gas fields are characteristic of the erosive protrusions of the Jurassic Period and occur occasionally in Cretaceous sediments of the Cenomanian, overlain by poorly permeable Miocene sediments. Proterozoic, Paleozoic, Mesozoic, and Cenozoic sediments are involved in the structure of the Bilche–Volytska zone.

The depression comprises two different layers—basement and sedimentary cover. The basement comprises Proterozoic–Paleozoic formations, while the sedimentary cover comprises Mesozoic–Cenozoic formations (Figure 8). The area of the Bilche–Volytska zone, which was dry for an extended time during the Cretaceous Period, was intensively eroded.

The Neogene sediments overlay Pre-Neogene sediments of varying ages, ranging from Proterozoic to Cretaceous [21].

| Era | Period | Epoch | Age | Suite name in Ukraine | Reservoir Horizon | Petrographic characteristics of rocks | Thickness, m |
|---|---|---|---|---|---|---|---|
| Cenozoic | Neogene (N) | Miocene $(N_1)$ | Messinian $(N_1mes)$ | $N_1st$ – Stebnytska $N_1s$ – Sarmatian Dashavska: $N_1dš_2$ – upper sub-suite $N_1dš_1$ – lower sub-suite | Horizon ND-5 (IV) Horizon ND-7 (V) Horizon ND-8 Horizon ND-9 | Thick monotonous layer of gray calcareous micaceous clays, siltstones, and sandstones with layers of tuffs and tuffites – molasses formation. | 0–4000 |
| | | | Tortonian $(N_1tor)$ | $N_1vr$ – Vorotyshchenska $N_1kos$ – Kosivska | Kosivska Suite | Layering of clays, siltstones, sandstones, and marls. | 100–1900 |
| | | | Serravallian $(N_1srv)$ | $N1tr$ – Tyraska | Horizon XVI | | 60–100 |
| | | | Langian $(N_1lan)$ | $N_1b$ – Badenian $N_1bg$ – Bogorodchanska | | Gypsum-anhydrite horizon. | 50–100 |
| Mesozoic | Cretaceous (K) | Upper $(K_2)$ | Maastrichtian | Senonian | | Sandstones, siltstones, argillites, marls, sandy limestones. | 0–100 |
| | | | Santonian | | | | 0–450 |
| | | | Coniacian | | | | 0–53 |
| | | | Turonian | | | | 0–123 |
| | | | Cenomanian | | | | 0–60 |
| | | Lower $(K_1)$ | Albian | | | Limestones, and calcareous argillites. | 0–210 |
| | Jurassic (J) | Upper $(J_3)$ | Tithonian Kimmeridgian | Nyzhnivska | | Limestones, dolomites with layers of marls, argillites, and sandstones. | 150–500 |
| | | | Oxfordian | Rava – Ruska | | | 150–400 |
| | | Middle $(J_2)$ | Callovian | Rudkivska Yavorivska | | Limestones, sandstones, siltstones, gravelites. | 0–60 |
| | | | Bathonian Bajocian | Kohanivska | | Argillites, siltstones, and sandstones. | 0–250 |
| | | Lower $(J_1)$ | Pliensbachian | Medynytska | | Sandstones, and argillites with layers of coal. | 200–500 |
| Paleozoic | Devonian (D) | Lower $(D_1)$ | Emsian Pragian Lochkovian | Dnistrovska $(D_1dn)$ | | Layering of sandstones, siltstones, and argillites. | 0–800 |
| | Silurian (S) | Wenlock | Homerian Sheinwoodian | Clay–carbonate layer $(S_{1-2}gk)$ | | Argillites, rarely siltstones, sandstones, and limestones. The carbonate increases up the section. | 0–1400 |
| | | Landovery | Telychian Aeronian Rhuddanian | | | | |
| | Ordovician (O) | Upper $(O_3)$ | Hirnantian Katian Sandbian | Molodova series $(O_{2-3}ml)$ | | Graptolitic argillite. | 0–150 |
| | | Middle $(O_2)$ | Darriwillian Dapingian | | | | |
| | Cambrian (€) | Series 2 | Stage 4 Stage 3 | Berezhkiv series $(€_{1-2}br)$ | | Layering of sandstones, siltstones, and argillites. | 0–1500 |
| | | Terreneuvian | Stage 2 Fortunian | Baltic series $(€_1bl)$ | | | |
| Proterozoic | | | | | | Shales, and phyllites with interlayers of sandstones, siltstones, and argillites. | >2000 |

**Figure 8.** Stratigraphic section of the Bilche–Volytska zone [21].

The UGS facilities in the western Ukraine complex are interconnected by a system of gas pipelines, which creates favourable conditions for the redistribution of gas flows to meet the needs of both local and distant consumers.

The achieved capacity of the complex in terms of the working gas volume is about 81.4% of the total amount of working gas in the country's gas storage facilities. The inventory of the production wells corresponds to 53% of the total number of production wells drilled in UGS facilities in Ukraine.

The western Ukrainian UGS complex is the most efficient gas storage complex in Ukraine, meeting the needs of the country's western region both in terms of the required gas storage volume and productivity. It ensures the security of gas supply not only in the

eastern region but also for transit supplies of export gas to western and eastern Europe. At the same time, there is a considerable shortage of UGS facilities in other regions of Ukraine (northern, central, eastern, and southern). This applies to the eastern and the Dnieper region, where the country's greatest industrial activity is located.

The Neogene sandstones confined to depleted gas reservoirs are grey with greenish and brownish tones. They are fine- and medium-grained, porous, and contain quartz and glauconite with carbonate and carbonate-clay cement, weakly cemented. The siltstones, mainly comprising quartz, are grey and dark grey with a greenish or brownish tint.

The Bilche–Volytsko–Uherske UGS facility lies within the $N_1srv$–$K_2$ reservoir. It is the largest store not only in Ukraine but in the whole of Europe and can contain more than 17 billion $m^3$ of natural gas [22].

The Upper Cretaceous ($K_2$) reservoirs comprise sandstones with layers of siltstones, calcareous argillites, marl layers, and pelitomorphic limestones. The sandstones are overlain by a reliable caprock of Langian ($N_1lan$) gypsum–anhydrite horizon. In this region, regional faults have considerable lengths, large amplitudes, depths, and durations of development.

## 2.2. Central Region

The central UGS complex is located within the Dnieper–Donets Basin (Figure 9). The Dnieper–Donets Basin lies almost entirely in Ukraine and is the main producer of hydrocarbons. The basin is bounded by the Voronezh High of the Russian Craton to the northeast and by the Ukrainian Shield to the southwest. The basin essentially consists of a Late Devonian rift overlain by clastic marine and alluvial deltaic sediments deposited in a Carboniferous to Early Permian postrift sag.

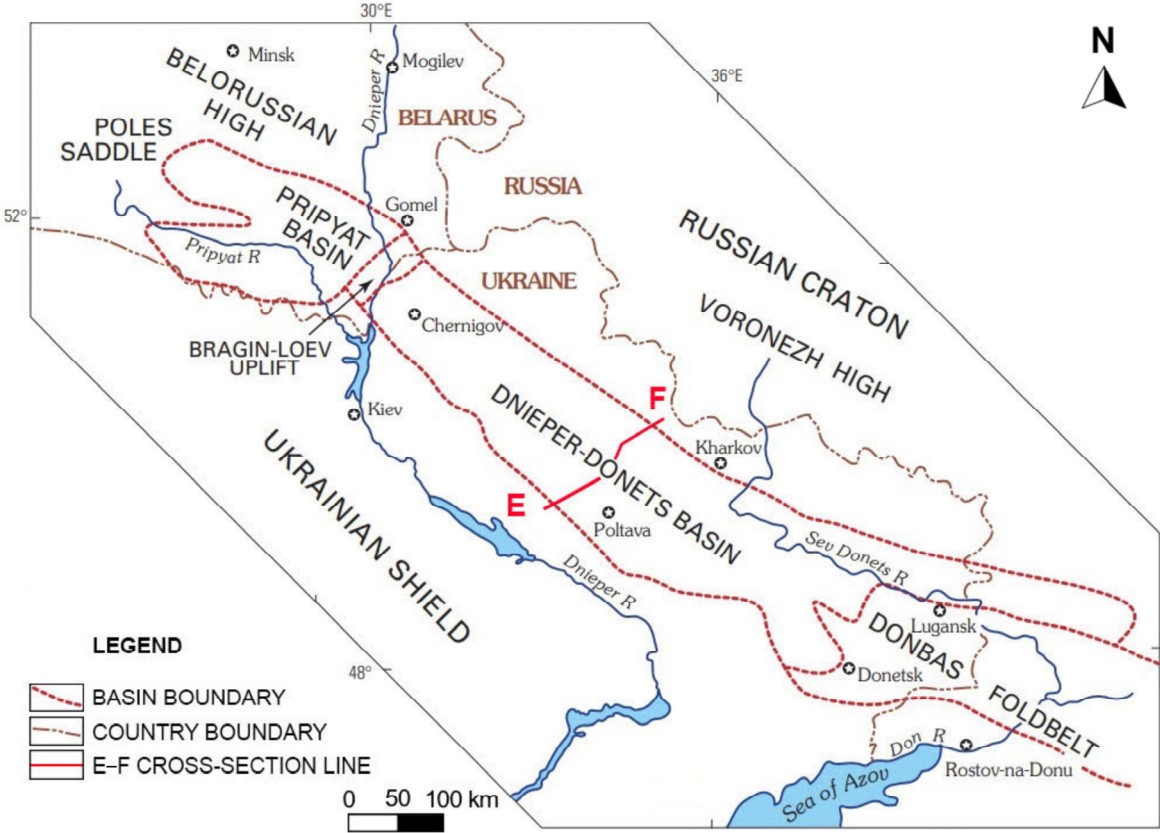

**Figure 9.** Location of Dnieper–Donets Basin. The indicated cross-section E–F is presented in Figure 10 (modified from [23]).

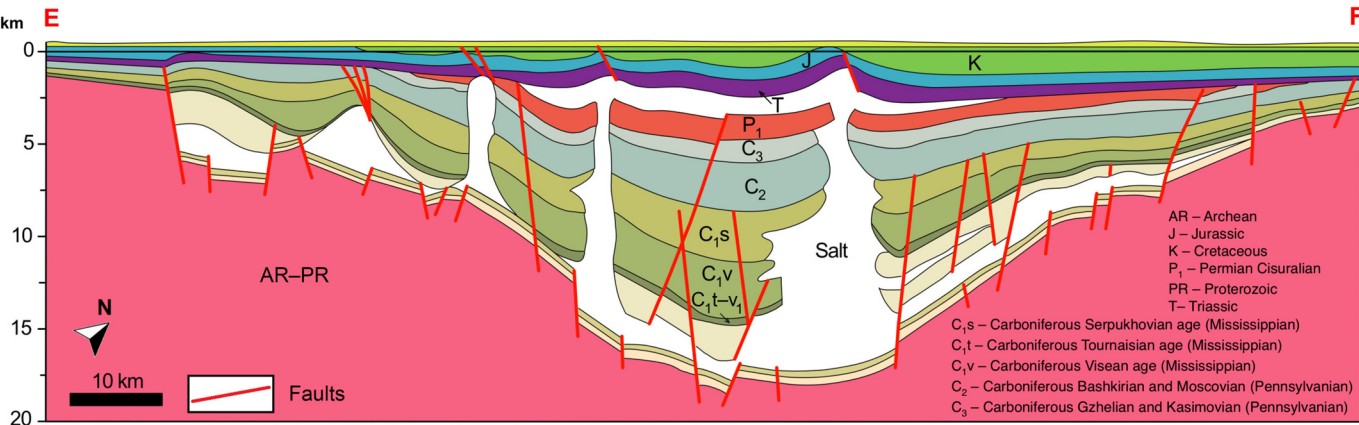

**Figure 10.** Regional geological cross-section E–F across the Dnieper–Donets Basin (modified from [23]). Location is shown in Figure 9.

The Devonian rift structure extends northwest to the Pripyat Basin in Belarus. The two basins are separated by the Bragin–Loev uplift, which serves as a Devonian volcanic centre. To the southeast, the Dnieper–Donets Basin has a gradual boundary with the Donbas fold belt, an area that has undergone structural inversion and deformation. The sedimentary succession of the basin comprises four tectono-stratigraphic sequences (Figures 9–11).

| System / Series / Stage | | | Lithology | Reservoir Horizon | Maximum thickness (m) | Sequence |
|---|---|---|---|---|---|---|
| Quaternary | | | | | 700 | Postrif platform |
| Cretaceous | | | | | 950 | |
| Jurassic | | | | Lower Bathonian Bajocian aquifer | 650 | |
| Triassic | | | | Lower Serebryanska | 900 | |
| Permian | Cisuralian | Kungurian | | | 1400 | |
| | | Artinskian | | | | |
| | | Sakmarian | | Under-Bryantsivskiy | | |
| | | Asselian | | | | |
| Carboniferous | Pensylvanian | Upper — Gzhelian | | | 1500 | Postrift sag |
| | | Kasimovian | | | | |
| | | Middle — Moscovian | | Horizon M-7 | 1200 | |
| | | Lower — Bashkirian | | Bashkirian Stage Horizon B-5+B-9 | 1200 | |
| | Missisipian | Upper — Serpukhovian | | | 800 | |
| | | Middle — Visean | | | 1700 | |
| | | Lower — Tournaisian | | | 750 | |
| Devonian | Upper | Famennian | | | 3600 | Synrift |
| | | Frasnian | | | 2000 | |
| | Middle | Givetian | | | 180 | Prerift platform |
| | | Eifelian | | | | |

**LEGEND**

Sand, mudstone　Volcanics
Sandstone, sand　Coal
Carbonate rock　Principal unconformity
Salt, anhydrite

**Figure 11.** Stratigraphic section of Dnieper–Donets Basin (modified from [23]).

The prerift platform sequence comprises Middle Devonian to Lower Frasnian clastic rocks deposited in a large intracratonic basin. The Upper Devonian synrift sequence is approximately 4–5 km thick and consists of marine carbonate, clastic, and volcanic rocks, along with two salt formations. These formations are deformed into salt domes and plugs, as shown in Figure 9.

The postrift sag sequence of the basin comprises Carboniferous and Lower Permian clastic marine and alluvial deltaic rocks, reaching thicknesses of up to 11 km in the southeastern part.

The Lower Permian interval includes a salt formation that is an important regional seal for oil and gas fields. The basin was strongly compressed in Artinskian (Early Permian) time and the southeastern basin areas were uplifted and deeply eroded, forming the Donbas fold belt. The postrift platform sequence includes Triassic to Tertiary rocks deposited in a shallow platform depression that extended well beyond the boundaries of the Dnieper–Donets Basin [24].

There are six oil- and gas-bearing stratigraphic units (complexes) in the Dnieper–Donets Basin with similar formations containing UGS reservoirs:

1. Mesozoic (Olyshivske, Chervonopartyzanske, Solokhivske, and Krasnopopivsle UGS facilities).
2. Upper Carboniferous–Lower Permian (Kehychivske UGS facility).
3. Middle Carboniferous (Verhunske and Proletarske UGS facilties).
4. Lower Carboniferous.
5. Devonian.
6. Precambrian.

The central UGS complex includes

6. Olyshivske (injection has not been carried out since 2012).
7. Chervonopartyzanske.
8. Solokhivske.
9. Kehychivske.

The UGS reservoirs are confined to the Mesozoic and Upper Carboniferous–Lower Permian oil and gas complexes. The Mesozoic sequence comprises depleted Jurassic strata and is the main component of the underground gas reservoirs of the eastern region (Olyshivske, Chervonopartyzanske, and Solokhivske). The Middle Jurassic ($J_2$) strata comprise clays with layers of sandstones containing shells and lignite lenses. The reservoir strata are sandstones. The Bathonian ($J_2$)-age clay formations are caprocks.

The Upper Carboniferous and Lower Permian sediments in the eastern region are represented by terrigenous and salt-water deposits. The difference between the salt-water formations is a sharp increase (up to 70%) in the proportion of evaporites (halite and anhydrite). The reservoirs comprise porous sandstones and fissured–cavernous anhydrites and limestones. The youngest Permian stratum comprises a sulphate–halogen layer, which serves as a first-class seal for the Permian reservoirs. Lithologically, it is rock salt with layers of anhydrites, saline siltstones, clays and sandstones, and potassium and magnesium salts.

The central UGS complex was built connected to the Kyiv system of main gas pipelines to ensure a reliable gas supply to consumers in the Kyiv, Khmelnytsky, Vinnytsia, Zhytomyr, Kirovohrad, Cherkasy, Chernihiv, Poltava, Sumy, and Kharkiv regions. The gas storage facilities are interconnected by a system of gas pipelines, which makes it possible to regulate the volume of injection and selection within the complex as needed.

The achieved volume of working gas within the complex is 12.2% of the total volume of working gas in the country's gas storage facilities. The inventory of production wells is 16% of the total number of production wells drilled in UGS facilities.

### 2.3. Eastern Region

The eastern UGS complex comprises (Figure 3) the following facilities in the Luhansk region:

10. Krasnopopivske.
11. Verhunske gas storage facilities (the injection has not been carried out since 2012).

The eastern UGS complex is in the Dnieper–Donets Basin. The UGS reservoirs are confined to the Mesozoic and Middle Carboniferous oil and gas complexes. The Mesozoic sequence includes the depleted Triassic gas field (Krasnopopivske). The Triassic is represented by the stratification of thin sand–clay layers of the Serebrianska suite ($T_2$). The reservoirs are sandstones. The Lower Triassic clay formations are the caprocks.

The UGS reservoirs in the Middle Carboniferous strata are confined to the Bashkirian stage (Verhunske). The lower part of the Bashkirian stack (Bashkirian plate) comprises sandy-clay sediments covered by a marine clay–carbonate layer, and the upper part comprises a caprock of sandstones and clays with carbonate and coal layers. The reservoir rocks comprise sandstones, and the seal is formed by a clay layer.

The eastern UGS complex was established in the Donetsk gas pipeline system to ensure a reliable gas supply to consumers in the Donbas. The achieved amount of working gas within the complex is 2.6% of the total amount in the country's UGS facilities. The stock of production wells is 8% of the total number of production wells drilled in UGS facilities.

### 2.4. Southern Region

The southern UGS complex is being built in the Dnieper region and in the Crimea as a system of gas pipelines that transports towards the Balkans and includes two UGS facilities:

12. Proletarske.
13. Hlibivske.

The UGS facilities in the southern UGS complex are in the Dnieper–Donets Basin (Proletarske) and the Prychornomorska Depression (Hlibivske). These UGS facilities are limited to the southern oil and gas region. The Proletarske UGS reservoir is confined to the Moscovian strata and Bashkirian stages of the Middle Carboniferous strata. The Moscovian strata comprise sandstones and mudstones and occasional limestones. The lower part of the Bashkirian (Bashkirian Plate) contains a pack of sandy-clay sediments covered by a marine clay–carbonate layer, and the upper part comprises a caprock of sandstones and clays with carbonate and coal layers.

The rocks of the UGS reservoir are predominantly sandstones and, to a lesser extent, fractured limestones. According to their evolutionary characteristics, the sandstones belong to layered, lenticular, bar-, channel-, and delta-deposited sandstones.

The Middle Carboniferous caprock is represented by a clay layer, mainly of lagoonal origin. The Hlibivske UGS reservoir is limited to the oil- and gas-bearing complex of the Prychornomorska Depression in strata of the Paleocene–Eocene age. The Prychornomorska Depression (Figure 12) is a geological structure extending from northwest to southeast in the zone where the East European platform is connected to the Scythian platform. The depression was formed by the long-term submergence of the southern slopes of the Ukrainian shield, which was most intense in the late Mesozoic and Cenozoic eras.

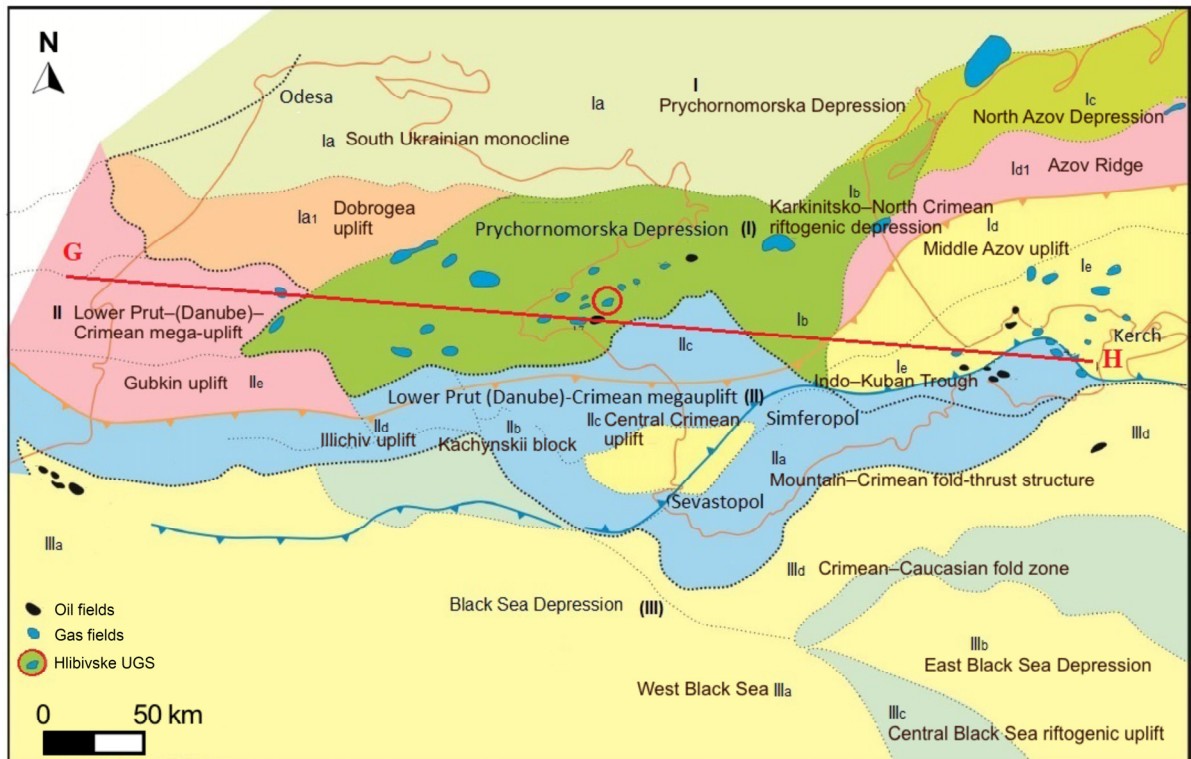

**Figure 12.** Tectonic map of the Azov–Black Sea region in southern Ukraine (modified from [25]). G–H line: a seismic–geological cross-section of the sedimentary cover is shown in Figure 13.

Latitudinal and meridional faults determined the block structure of the crystalline basement. The Prychornomorska Depression's geological structure comprises Palaeozoic, Mesozoic, and Cenozoic sediments, which can be up to 6000–8000 m thick (Figures 13 and 14) [26]. These sedimentary rocks are related to the primary hydrocarbon regions and comprise various geological formations.

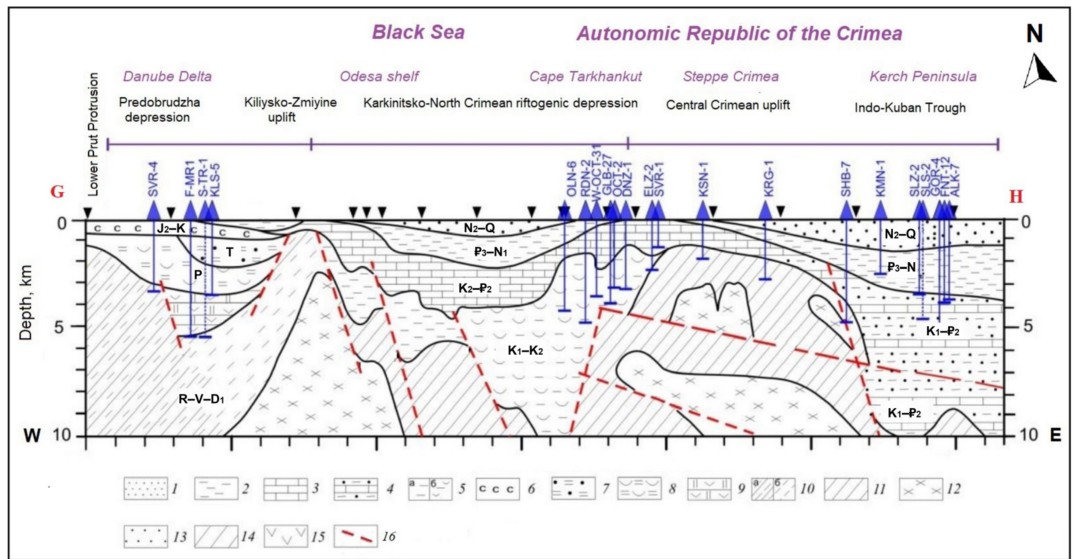

**Figure 13.** Seismic–geological cross-section of the sedimentary cover in southern Ukraine with probable fault zones along the G–H line that is shown in Figure 12 (modified from [27]): 1—middle Miocene–Quaternary ($N_2$–Q) sand and mudstone; 2—Eocene–Oligocene ($P_2$–$P_3$) and Oligocene–Lower Miocene ($P_3$–$N_1$) sediments; 3—carbonate rocks of Upper Cretaceous–Eocene ($K_2$–$P_2$); 4—Lower Cretaceous–

Eocene ($K_1$–$P_2$) siltstones and marls; 5—terrigenous (a) and terrigenous–volcanogenic (b) rocks; 6—Middle Jurassic–Lower Cretaceous ($J_2$–K) terrigenous–clay layer; 7—Triassic (T) argillites and anhydrites; 8—Permian (P) argillites and clays; 9—Middle Devonian–Carboniferous ($D_2$–C) limestones and dolomites; 10—Riphean Lower Devonian undissected platform cover complex (R–V–$D_1$); 11—Paleozoic Mesozoic basement; 12—Doryphean granite–gneiss layer; 13—sedimentary cover; 14—Paleozoimesozoic basement; 15—volcanogenic complex; 16—faults.

| Era | Period | Epoch | | Age | Reservoir Horizon | Petrographic characteristics of rocks | Thickness, m |
|---|---|---|---|---|---|---|---|
| Cenozoic | Neogene (N) | Pliocene (N₂) | | Piacenzian | | Calcareous clays, sands, and sandstones. | 0–200 |
| | | | | Zanclean | | Shell, and oolitic limestones. | 0–20 |
| | | Miocene (N₁) | | Messinian | | Marl–clay, and sandy-silt rocks, oolitic limestones. | 0–100 |
| | | | | Tortonian | | | 0–420 |
| | | | | Serravallian | | Sandy–clay shell limestones, marls. | 0–750 |
| | | | | Langhian | | | 0–2000 |
| | | | | Burdigalian | | A uniform layer of gray, and dark gray slightly calcareous clays, occasionally with layers of siltstones, sandstones. | |
| | | | | Aquitanian | | | |
| | Paleogene (P) | Oligocene (P₃) | | Chattian | | | 0–1750 |
| | | | | Rupelian | | | |
| | | Eocene (P₂) | | Priabonian | | Calcareous clays, marls, and argillaceous limestones. | 0–410 |
| | | | | Bartonian | | | 0–100 |
| | | | | Lutetian | | | 0–400 |
| | | | | Ypresian | | | 0–120 |
| | | Paleocene (P₁) | | Thanetian | | Fine–grained argillaceous limestones, and blue–gray marls, less often clays, and sandstones. | 0–200 |
| | | | | Selandian | Brachyform fold | | 0–200 |
| | | | | Danian | | | 0–200 |
| Mezozoic | Cretaceous (K) | Upper (K₂) | | Maastrichtian | | Mostly carbonate rocks: limestones, dolomites. | 0–920 |
| | | | | Campanian | | | 0–430 |
| | | | | Santonian | | | 0–320 |
| | | | | Coniacian | | | |
| | | | | Turonian | | | 0–950 |
| | | | | Cenomanian | | | 0–600 |
| | | Lower (K₁) | | Albian | | Sedimentary–volcanogenic layer. | 0–1100 |
| | | | | Aptian | | Silty clays, and argillites. | 0–240 |
| | | | | Barremian | | Variegated continental sandstones, gravels, and clays. | |
| | | | | Hauterivian | | | |
| | Jurassic (J) | Upper (J₃) | | Tithonian | | | >1500 |
| | | | | Kimmeridgian | | Carbonate–evaporite complex: dolomites, limestones, gypsum. | |
| | | | | Oxfordian | | Organic–rich limestones. | |
| | | Middle (J₂) | | Callovian | | Terrigenous–clay layer of argillites, siltstones. | 210–3000 |
| | | | | Bathonian | | | |
| | | | | Bajocian | | | |
| | Triassic (T) | Upper (T₃) | | Norian | | Limestones with layers of argillites, and anhydrites. | 80–640 |
| | | | | Carnian | | | |
| | | Middle (T₂) | | Ladinian | | Siltstones, sandstones, and mudstones. | |
| | | | | Anisian | | | |
| | | Lower (T₁) | | Olenekian | | Coarse–clastic terrigenous rocks. | >200 |
| Paleozoic | Permian (P) | Cisuralian | | Kungurian | | Argillites, siltstones, and red sandstones with layers of conglomerates, and effusive. | |
| | | | | Artinskian | | | |
| | | | | Sakmarian | | | |
| | | | | Asselian | | | |
| | Carboniferous (C) | Mississipian | Upper | Serpukhovian | | Terrigenous clay stratum of gray mudstones, siltstones, and sandstones with layers of coal. | <650 |
| | | | Middle | Visean | | Limestones with layers of siltstones, and sandstones. | 350 |
| | | | Lower | Tournaisian | | Dolomites, and anhydrites with layers of dolomitized limestones, and argillites. | 500 |
| | Devonian (D) | Upper (D₃) | | Famennian | | Dolomites, limestones, and anhydrites. | <850 |
| | | | | Frasnian | | | |
| | | Middle (D₂) | | Givetian | | Limestones, and dolomites, anhydrites with layers of argillites, and marls. | 600–850 |
| | | | | Eifelian | | | |
| | | Lower (D₁) | | Emsian | | Calcareous argillites, and siltstones with thin layers of marls, tuff sandstones, dolomites, limestones. | >2000 |
| | | | | Pragian | | | |
| | | | | Lochkovian | | | |
| | Silurian (S) | Pridoli | | | | Limestones, and dolomotic. | >650 |
| | | Ludlow | | Ludfordian | | Dolomitic marls. | |
| | | | | Gorstian | | | |
| | | Wenlock | | Homerian | | Limestones. | |
| | | | | Sheinwoodian | | | |
| | | Landovery | | Telychian | | Limestones. | |
| | | | | Aeronian | | | |
| | | | | Rhuddanian | | | |
| | Ordovician (O) | Middle (O₂) | | Darriwillian | | Gray quartz calcareous sandstones, and limestones. | 500 |
| | | | | Dapingian | | | |
| | Cambrian (Є) | Terreneuvian | | Stage 2 | | Different–grained sandstones, siltstones, argillites, gravelites. | 300 |
| | | | | Fortunian | | | |
| | Neo-proterozoic | | | | | Terrigenous layer of mudstones, siltstones, and sandstones. | >300 |
| | Paleo-proterozoic | | | | | Igneous, and metamorphic rocks: migmatites, gneisses, granodiorites. | |

**Figure 14.** Combined stratigraphic section of the southern oil and gas region of Ukraine—the Crimean Plain, the Northern Black Sea Coast, and the Northwestern Black Sea Shelf (modified from [26]).

Large uplifts and troughs are complicated by local structures that are often oil- and gas-bearing. Oil and gas are found in rocks from the Neogene to the Devonian age, but gas and gas condensate fields have only been found in the Paleogene and Lower Cretaceous strata at depths of 350 to 4500 m. They are mainly associated with vaulted parts of anticlinal folds. The hydrocarbon reservoirs comprise sandstones, siltstones, and organic-detrital limestones.

The Hlibivske UGS facility, located in the Prychornomorska Depression, includes Palaeocene sediments of fine-grained clayey limestones and marls. The Lower Palaeocene comprises limestones and marls, while the upper comprises carbonate rocks. Dense calcareous clays serve as caprocks.

This facility ensures a consistent gas supply for domestic users, with additional gas transport to Moldova, the Balkan Peninsula, and Turkey through the southern regions of Ukraine. The complex currently holds 3.8% of the total working gas supply. This complex holds 23% of all the production wells drilled in Ukraine's UGS facilities.

## 3. Data and Methods

For Ukraine, the Hystories data collection exercise built on the primary reservoir data collected as part of the ESTMAP (Energy Storage Mapping And Planning) project [28]. All the UGS facility data collated in the ESTMAP project were checked and updated during the Hystories project. Data on all the analysed UGS facilities were collected from scientific publications and reports [29–39] available in the State Research and Development Enterprise "Geoinform of Ukraine". In addition, the comprehensive six-volume edition of the "Atlas of Oil and Gas Fields of Ukraine" (1998–1999), which contains detailed information on all oil- and gas-bearing regions of Ukraine, as well as oil, gas, and condensate fields, was used.

The reports from Geoinform of Ukraine, which outline the geological properties of UGS facilities, were mainly used to identify appropriate sites for $H_2$ or $CO_2$ storage. These reports provide valuable information, including seismic data, details about wells, core samples, and reservoir models for the relevant fields. In addition, over 50 research publications, papers, and open data from energy company websites were analysed and included in the Hystories study of Ukraine.

A high-level estimate of the range of the $H_2$ storage potential can be estimated by considering the volumetric characteristics of UGS reservoirs. In UGS facilities, the three relevant volumes for this study are the physically unrecoverable gas, the cushion gas, and the working gas capacity. Unrecoverable gas is the volume that is trapped in the pore space and cannot be withdrawn. The cushion gas is used to maintain reservoir pressure and ensure deliverability. The working gas capacity is the gas which is injected and withdrawn to meet demand. This study considers two scenarios: (1) the UGS reservoir would be depleted such that only the physically unrecoverable gas remains and hydrogen is used as the cushion gas, and (2) the UGS facility is converted to pure $CO_2$ storage. The volumetric capacity ranges are obtained by combining the reported working gas volume in the UGS facilities and $H_2$ density in situ reservoir conditions (temperature and pressure properties, Table 1).

**Table 1.** Ukraine's underground natural gas storage system (modified from [16,22,40]).

| | UGS Facility Name | Year | Gas Volume, Mm$^3$ | | Number of Production Wells | Reservoir Type | Region |
|---|---|---|---|---|---|---|---|
| | | | Total Gas Volume (Including Cushion) | Working Gas Volume | | | |
| 1 | Uherske | 1969 | 3850 | 1900 | 88 | Depleted deposit | Western |
| 2 | Bilche–Volytsko–Uherske | 1983 | 33,450 | 17,050 | 341 | Depleted deposit | Western |
| 3 | Oparske | 1979 | 4570 | 1920 | 76 | Depleted deposit | Western |
| 4 | Dashavske | 1973 | 5265 | 2150 | 100 | Depleted deposit | Western |
| 5 | Bohorodchanske | 1979 | 3420 | 2300 | 156 | Depleted deposit | Western |
| 6 | Olyshivske | 1964 | 660 | 310 | 40 | Aquifer | Central |
| 7 | Chervonopartyzanske | 1968 | 2973.8 | 1500 | 67 | Aquifer | Central |
| 8 | Solokhivske | 1987 | 2100 | 1300 | 81 | Depleted deposit | Central |
| 9 | Kehychivske | 1986 | 1300 | 700 | 53 | Depleted deposit | Central |
| 10 | Krasnopopivske | 1973 | 800 | 420 | 40 | Depleted deposit | Eastern |
| 11 | Verhunske | 1975 | 951 | 400 | 73 | Depleted deposit | Eastern |
| 12 | Proletarske | 1986 | 2980.3 | 1000 | 251 | Depleted deposit | Southern |
| 13 | Hlibivske | 1983 | 1881.1 | 1000 | 84 | Depleted deposit | Southern |
| | Total | | 56,366 | 31,950 | 1450 | | |

The $H_2$ capacity in $M_{H_2}$ (cm$^3$) of the UGS was estimated with different levels of confidence. Using the total volume of the gas reservoir, $V_{total}$ (Table 2), the total $H_2$ storage capacity ($M_{H_2 total}$) was estimated:

$$M_{H_2 total} = V_{total} \times \rho_{H_2 r} \tag{1}$$

The $H_2$ density in in situ reservoir conditions ($\rho_{H_2 r}$) was calculated as a function of pressure and temperature using models for thermodynamic properties of pure fluids [41]. The operating mode of the UGS facilities is determined by the maximum and minimum pressure of the reservoir.

The operating mode of the UGS facilities is determined by the maximum and minimum pressure of the reservoir. The maximum pressure corresponds to the state when the UGS facility is completely filled with gas, while the minimum pressure corresponds to the state when only cushion gas is in the reservoir. There is also an allowable (permissible) maximum gas pressure in UGS facilities, which depends on geological characteristics and technical factors. Typically, the maximum allowable pressure is close to or slightly below the original reservoir pressure before depletion. It is also worth noting that $CO_2$ storage is usually conducted at depths below 800 m where the $CO_2$ is a supercritical fluid. $H_2$ storage and UGS reservoirs can be shallower and, therefore, this brings an additional consideration for the conversion of UGS facilities for $CO_2$ storage.

**Table 2.** Geological and technical parameters of UGS facilities in Ukraine.

| UGS | Formation Name | Age | Trap Name in Ukraine | UGS Total Area km$^2$ | Temp., °C | Pressure, MPa Avg (Min–Max) | Permeability, Md ($10^{-15}$ m$^2$) Avg (Min–Max) | Porosity, % Avg (Min–Max) | Top Reservoir Depth, m | Thickness, m Avg (Min–Max) | Reservoir Lithology | Seal Lithology |
|---|---|---|---|---|---|---|---|---|---|---|---|---|
| 1. Uherske | Molasse | N$_1$mes | Horizon ND-8 | 21.5 | 30 | 5.37 | 30 (12–47) | 20 | 687.5 | 70 | Sandstone | Anhydrite |
| | Molasse | N$_1$mes | Horizon ND-9 | 4.95 | 30 | 5.5 | 39 | 20 | 750 | 45 | Sandstone | Anhydrite |
| 2. Bilche–Volytsko–Uherske | Flysch | N$_1$srv-K$_2$ | Horizon XVI | 74 | 42 | 5.02 (2.4–7.7) | 20 (19.3–60.5) | 25 | 965 | 187 | Sandstone | Anhydrite |
| 3. Oparske | Molasse | N$_1$mes | Horizon ND-5 (IV) | 19.4 | 27 | 5.4 (2.6–8.3) | 103 (75–131) | 27 | 620 | 23 | Sandstone | Claystone |
| | Molasse | N$_1$mes | Horizon ND-7 (V) | 10.3 | 27 | 5.4 (2.6–8.3) | 103 (75–131) | 15 | 715 | 20.6 | Sandstone | Claystone |
| | Molasse | N$_1$mes | Horizon ND-8 (VI) | 24 | 27 | 5.4 (2.6–8.3) | 103 (75–131) | 10 | 820 | 20.4 | Sandstone | Claystone |
| 4. Dashavske | Molasse | N$_1$mes | Horizon ND-8 | 34.4 | 25 | 3.8 (1.9–5.8) | 20–1200 | 26 | 657 | 9 | Sandstone | Siltstones, Clays |
| | Molasse | N$_1$mes | Horizon ND-9 | 5.49 | 25 | 3.8 (1.9–5.8) | 20–1200 | 25 | 710 | 15 | Sandstone | Siltstones, Clays |
| 5. Bohorodchanske | Molasse | N$_1$tor | Kosivska Suite | 14 | 44 | 6.7 (3–10.5) | 3 | 15 | 1130 | 150 | Sandstone | Claystone |
| 6. Olyshivske | Sandy clay | J$_2$b-bt | Bathonian-Bajocian aquifer | 24.12 | 25 | 5.6 | 2 | 35 | 560.5 | 20 | Sandstone | Claystone |

**Table 2.** *Cont.*

| UGS | Formation Name | Age | Trap Name in Ukraine | UGS Total Area km² | Temp., °C | Pressure, MPa Avg (Min–Max) | Permeability, Md (10⁻¹⁵ m²) Avg (Min–Max) | Porosity, % Avg (Min–Max) | Top Reservoir Depth, m | Thickness, m Avg (Min–Max) | Reservoir Lithology | Seal Lithology |
|---|---|---|---|---|---|---|---|---|---|---|---|---|
| 7. Chervonopar-tyzanske | Sandy clay | $J_2bt$ | Lower Bathonian | 13 | 25 | 4.5 | 2 | 30 | 536.5 | 10–20 | Sandstone | Claystone |
| | Sandy clay | $J_2b$ | Bajocian aquifer | 13 | 25 | 3.8 | 2 | 30 | 544 | 10–16 | Sandstone | Claystone |
| 8. Solokhivske | Sandy clay | $J_2b$ | Bajocian horizon | 57.6 | 34 | 7.9 | 3 | 25 | 903.5 | 67.5 | Sandstone | Claystone |
| 9. Kehychivske | Saline | $P_1s$ | Under-Bryantsivskiy horizon | 10.9 | 55 | 15.8 | 8 | 14 (6–23) | 2060 | 5.2 | Sandstone | Salt |
| 10. Krasnopopivske | Sandy clay | $T_1$ | Lower Sere-bryanska Sub-Suite | 12 | 22 | 3.5 | 1.1 | 19 | 455 | 8 | Sandstone | Claystone |
| 11. Verhunske | Terrigenous carbonate | $C_2b$ | Bashkirian Stage | 12 | 30 | 11.8 (8.2–15.4) | 0.6 | 22 | 1224 | 9.7 | Sandstone | Claystone |
| 12. Proletarske | Terrigenous carbonate | $C_2m$ | Horizon M-7 | 13 | 35 | 8.7 (6.0–11.4) | 104 (63–145) | 20 (11–28) | 1440 | 5–26.1 | Sandstone | Claystone |
| | Terrigenous carbonate | $C_2b$ | Horizon B-5+B-9 | 13 | 35 | 8.7 (6.0–11.4) | (0.2–474) | 20 (12–25) | 1785 | 6.6–66.2 | Sandstone | Claystone |
| 13. Hlibivske | Clay–carbonate | $P_1d$-sl | Brachyform fold | 9.4 | 68 | 11.2 | 0.5 | 19 | 1050 | 120 | Limestone | Claystone |
| Average (Min–Max) | | | | 18.2 (4.95–74) | 35 (22–68) | 6.7 (1.9–15.4) | 34.9 (0.2–1200) | 21.95 (6–28) | 964 (455–2151) | 44.8 (5–66) | | |

Avg—average.

The maximum values of reservoir pressure (Table 2) were used to estimate the $H_2$ density. The volume of the active zone of the UGS reservoir ($V_{active}$, Table 1) was used to estimate the active capacity of the storage site ($M_{H_2active}$), considering the reservoir in situ fluids and cushion gas that could be used during $H_2$ storage:

$$M_{H_2active} = V_{active} \times \rho_{H_2r} \tag{2}$$

The cushion gas volume of the potential $H_2$ storage site was estimated for two potential cases: (1) if the cushion gas is $H_2$ ($V_{H_2}cg$), the volume is estimated as the difference between the $M_{H_2total}$ and the $M_{H_2active}$ (Table 3) and (Table 2); (2) if the cushion gas is $CO_2$ ($V_{CO_2}cg$), the volume is estimated as the difference between the optimistic $CO_2$ storage capacity ($M_{CO_2}Opt.$) and the conservative approach ($M_{CO_2}Cons.$) (Table 4).

**Table 3.** The potential for the $H_2$ storage capacity, in Mt, in the studied UGS facilities in Ukraine.

| UGS Facility | Total Gas Volume Mm³ | Working Gas Volume Mm³ | Temp., °C | Pressure, MPa (max) | $\rho_{H_2r}$ (kg/m³) | $M_{H_2total}$ (Mt) | $M_{H_2active}$ (Mt) | $M_{H_2}cg$ (Mt) | $M_{H_2}$ Energy (TWh) |
|---|---|---|---|---|---|---|---|---|---|
| 1. Uherske | 3850 | 1900 | 30 | 8.5 | 6.46 | 24.9 | 12.3 | 12.6 | 4.8462 |
| 2. Bilche–Volytsko–Uherske | 33,450 | 17,050 | 42 | 10.38 | 7.47 | 250 | 127.4 | 122.6 | 50.1956 |
| 3. Oparske | 4570 | 1920 | 27 | 7.7 | 5.92 | 27.1 | 11.4 | 15.7 | 4.4916 |
| 4. Dashavske | 5265 | 2150 | 25 | 6.65 | 5.14 | 27.1 | 11.1 | 16.0 | 4.3734 |
| 5. Bohorodchanske | 3420 | 2300 | 44 | 10.2 | 7.36 | 25.2 | 16.6 | 8.2 | 6.5404 |
| 6. Olyshivske | 660 | 310 | 25 | 7 | 5.44 | 3.6 | 1.7 | 1.9 | 0.6698 |
| 7. Chervonopar-tyzanske | 2973.8 | 1500 | 25 | 5.3 | 4.17 | 12.4 | 6.3 | 6.1 | 2.4822 |
| 8. Solokhivske | 2100 | 1300 | 23 | 9.6 | 7.18 | 15.1 | 9.3 | 5.7 | 3.6642 |
| 9. Kehychivske | 1300 | 700 | 55 | 15.9 | 10.61 | 13.8 | 7.4 | 6.4 | 2.9156 |
| 10. Krasnopopivske | 800 | 420 | 22 | 5 | 3.99 | 3.2 | 1.7 | 1.5 | 0.6698 |
| 11. Verhunske | 951 | 400 | 30 | 12.4 | 9.12 | 8.7 | 3,6 | 5.0 | 1.4184 |
| 12. Proletarske | 2980.3 | 1000 | 35 | 16.5 | 11.63 | 34.7 | 11.6 | 23.0 | 4.5704 |
| 13. Hlibivske | 1881.2 | 1000 | 68 | 11.2 | 7.46 | 14.0 | 7.46 | 6.57 | 2.93924 |
| Average (Min–Max) | 4939 (660–33,450) | 2458 (310–17,050) | 35 (22–68) | 9.72 (5–16.5) | 7 (4–11.6) | 35.4 (3.2–250) | 17.5 (1.7–127.4) | 17.8 (1.5–122.6) | 7.4 (0.7–50.2) |
| Total | 64,201.2 | 31,950 | | | | 459.6 | 228.2 | 231.27 | 89.77684 |

Mt—millions of tonnes; $\rho_{H_2r}$—the density of $H_2$ in situ reservoir conditions; $M_{H_2total}$—the total storage capacity of $H_2$ in the reservoir, excluding cushion gas and in situ fluids; $M_{H_2active}$—the working or active storage capacity of $H_2$ in the reservoir, considering cushion gas and fluids in the reservoir; $M_{H_2}cg$—the volume of the $H_2$ cushion gas; $M_{H_2}$ Energy—the storage capacity of $H_2$ in energy units of Terawatt hours (TWh).

**Table 4.** The potential for the $CO_2$ storage capacity, in Mt, in the studied UGS facilities in Ukraine.

| UGS Facility | Total Gas Volume Mm³ | Working Gas Volume Mm³ | Temp., °C | Pressure, MPa (max) | $\rho_{CO_2r}$ (kg/m³) | $M_{CO_2}Opt.$ (Mt) | $M_{CO_2}Cons.$ (Mt) | $M_{CO_2}cg$ (Mt) | $CO_2$ State |
|---|---|---|---|---|---|---|---|---|---|
| 1. Uherske | 3850 | 1900 | 30 | 8.5 | 723 | 2783.6 | 1373.7 | 1409.9 | Fluid |
| 2. Bilche–Volytsko–Uherske | 33,450 | 17,050 | 42 | 10.38 | 593.85 | 19,864.3 | 10,125.1 | 9739.2 | SC fluid |
| 3. Oparske | 4570 | 1920 | 27 | 7.7 | 718 | 3281.3 | 1378.6 | 1902.7 | Fluid |
| 4. Dashavske | 5265 | 2150 | 25 | 6.65 | 728.7 | 3836.6 | 1566.7 | 2269.9 | Fluid |
| 5. Bohorodchanske | 3420 | 2300 | 44 | 10.2 | 539.7 | 1845.8 | 1241.3 | 604.5 | SC fluid |
| 6. Olyshivske | 660 | 310 | 25 | 7 | 700.95 | 462.6 | 217.3 | 245.3 | Fluid |
| 7. Chervonopar-tyzanske | 2973.8 | 1500 | 25 | 5.3 | 144.1 | 428.5 | 216.2 | 212.3 | Gas |
| 8. Solokhivske | 2100 | 1300 | 23 | 9.6 | 823.9 | 1730.2 | 1071.1 | 659.1 | Fluid |
| 9. Kehychivske | 1300 | 700 | 55 | 15.9 | 670.3 | 871.1 | 469.2 | 401.9 | SC fluid |

| UGS Facility | Total Gas Volume Mm$^3$ | Working Gas Volume Mm$^3$ | Temp., °C | Pressure, MPa (max) | $\rho_{CO_2r}$ (kg/m$^3$) | $M_{CO_2}$Opt. (Mt) | $M_{CO_2}$Cons. (Mt) | $M_{CO_2}$cg (Mt) | CO$_2$ State |
|---|---|---|---|---|---|---|---|---|---|
| 10. Krasnopopivske | 800 | 420 | 22 | 5 | 137.3 | 409.2 | 137.3 | 271.9 | Gas |
| 11. Verhunske | 951 | 400 | 30 | 12.4 | 807.7 | 646.2 | 339.2 | 307 | Fluid |
| 12. Proletarske | 2980.3 | 1000 | 35 | 16.5 | 829.13 | 788.5 | 331.7 | 456.8 | SC fluid |
| 13. Hlibivske | 1881.1 | 1000 | 68 | 11.2 | 327.5 | 616.1 | 327.5 | 288.6 | SC fluid |
| Average (Min–Max) | 5295 (660–33,450) | 2458 (310–17,050) | 35 (22–68) | 9.7 (5–16.5) | 596 (137–829) | 2890 (409–19,864) | 1446 (137–10,125) | 1444 (212–9739) | |
| Total | 64,201.2 | 31,950 | | | | 37,564 | 18,794.9 | 18,769.1 | |

Mt—millions of tonnes; $\rho_{CO_2r}$—density of CO$_2$ in situ reservoir conditions; $M_{CO_2}$Opt.—optimistic CO$_2$ storage capacity; $M_{CO_2}$Cons.—conservative CO$_2$ storage capacity; $M_{CO_2}$cg—volume of CO$_2$ cushion gas for H$_2$; CO$_2$ state—CO$_2$ state of aggregation in in situ conditions.

The theoretical CO$_2$ storage capacity of the structures was estimated using the same approach as for H$_2$. The optimistic approach, $M_{CO_2}$Opt., was estimated using the $V_{total}$, and for the $M_{CO_2}$Cons., the $V_{active}$ was used:

$$M_{CO_2}\text{Opt.} = V_{total} \times \rho_{CO_2r} \tag{3}$$

$$M_{CO_2}\text{Cons.} = V_{active} \times \rho_{CO_2r} \tag{4}$$

The CO$_2$ density in in situ reservoir conditions ($\rho_{CO_2r}$) value depends on the in situ reservoir pressure and temperature and was estimated using a function of the states of CO$_2$ under in situ conditions [42]. The maximum values of the reservoir pressure (Table 1) were used to estimate the CO$_2$ density. To calculate the active H$_2$ storage capacity in energy units, the heating value for H$_2$ of 39.4 kWh/kg was applied [43].

## 4. Potential for CO$_2$ and H$_2$ Storage

For this paper, the UGS data that were collected as part of the Hystories project were further reviewed and updated (Table 2) with additional details to assess the potential for H$_2$ and CO$_2$ storage in UGS facilities in Ukraine (Tables 3 and 4). The 13 assessed Ukrainian UGS facilities are represented by 12 sandstone and 1 limestone reservoir facilities. They are characterised by a wide variation of parameters, including the areas in the range of 5–74 km$^2$ (average 18.2 km$^2$), reservoir rock thicknesses in the range of 5–187 m (average 44.8 m), and reservoir top depths in the range of 455–2151 m (average 964 m). The Oparske UGS facility exploits three reservoir layers located at the depth of 620, 715 and 820 m, and four UGS facilities (Uherske, Dashavske, Chervonopartyzanske, and Proletarske) exploit two reservoir layers. The reservoir rocks are characterised by porosities in the range of 6–35%, with an average of 22% for all the reservoir layers. The reservoir temperatures increased from 22 °C at the depth of 455 m (Krasnopopivske) to 68 °C at the depth of 1050 m (Hlibivske), with an average of 35 °C in Proletarske at the depth of about 2 km (Table 2).

In the context of climate change mitigation efforts, CO$_2$ storage is an important technology for storing CO$_2$ emissions from industrial processes and power plants. UGS facilities in Ukraine can potentially serve as secure and large-scale CO$_2$ storage sites by repurposing existing sites for CO$_2$ storage. Additionally, there could be a potential for utilising the storage capacity in Ukraine for importing CO$_2$ from other sources outside Ukraine or for transferring CO$_2$ from one area of Ukraine to another.

Estimating the CO$_2$ storage capacity accurately is essential for quantifying the potential emission reductions and optimizing subsurface planning. This information is crucial for demonstrating the feasibility and effectiveness of CCUS projects in achieving emission

reduction targets. The analysis carried out during this study made it possible to estimate the potential of the $CO_2$ storage capacity.

Repurposing existing UGS facilities for $CO_2$ storage can be a cost-effective solution. This approach utilises existing infrastructure, reduces the need to build completely new $CO_2$ storage facilities, and uses well-understood reservoirs with the proven ability to store buoyant fluids. An $H_2$-specific evaluation of the sites, particularly the seals, would be required since $H_2$ can permeate through some materials that could trap $CO_2$ and natural gas.

In this study, the quantitative range of theoretical $H_2$ and $CO_2$ storage capacities of the 13 reported onshore UGS facilities in Ukraine was estimated for the first time (Tables 3 and 4). The amount of cushion gas required for $H_2$ storage was also calculated. Based on the specific in situ reservoir conditions, the density and state of $CO_2$ were estimated for each storage site.

The Bilche–Volytsko–Uherske storage site, with the largest area and average reservoir thickness (74 km$^2$ and 187 m, respectively), offers the largest storage capacity for both $CO_2$ and $H_2$ amongst all the structures analysed. The sandstone storage reservoir has excellent average porosity (21%) and permeability (1000–2000 mD). Among all the sites investigated, this storage site is the most suitable and feasible for both $H_2$ and $CO_2$ storage.

If the structure was depleted and $H_2$ was used as a cushion gas, the facility could potentially store over 127 Mt of $H_2$ in the working zone of the underground storage site (Equation (2)) and about 250 Mt of $H_2$, including cushion gas (Equation (1)).

If the facility were converted to pure $CO_2$ storage, the Bilche–Volytsko–Uherske site could potentially store almost 20 gigatonnes of $CO_2$ in an optimistic estimation, while a conservative option assumes a capacity of over 10 gigatonnes.

The $CO_2$ will be stored in the geological structure in a highly dense fluid state.

Although the Bilche–Volytsko–Uherske storage facility is considered the most promising, there is currently no public information regarding plans to repurpose it or other Ukrainian UGS facilities for $H_2$ or $CO_2$ storage. However, in June 2024, the Naftogaz Group, the subsidiary of which is Ukrtransgaz (a UGS operator), signed a memorandum of understanding with RAG Austria AG, Austria's largest gas storage operator, to collaborate on $H_2$ storage in gas reservoirs. This demonstrates intent not only to exchange expertise in $H_2$ storage in sandstone reservoirs but also to collaborate on the technical aspects of $H_2$ storage in gas facilities. This agreement showcases Ukraine's potential to become a key partner in the development of $H_2$ energy within the EU.

The optimistic $CO_2$ storage capacity of the 13 UGS facilities was estimated to be in the range of 0.4–19.9 Gt, and for all the analysed structures combined it was around 37.6 Gt. The "conservative" $CO_2$ storage capacity was in the range of 0.14–10.1 Gt, with a total capacity in all the structures of 18.8 Gt. The optimistic estimate represents the assumption that the full volume of working gas plus cushion gas can be replaced with $CO_2$ (see Equation (3)). The conservative estimate assumes that only the working volume of natural gas can be replaced with $CO_2$ for storage (see Equation (4)). The storage capacity of $CO_2$ cushion gas for $H_2$ storage is estimated to be in the range of 0.21–9.7 Gt, with a total of 18.8 Gt of $CO_2$.

The "total" $H_2$ storage volume of each of the 13 structures was estimated to be in the range of 3.2–250 Mt (average 35.4 Mt). The "total" $H_2$ storage volume of all the structures combined is approximately 459.6 Mt. The "working" $H_2$ storage volumes were estimated to be in the range of 1.7–127.4 Mt or 0.7–50.2 TWh (average 17.5 Mt or 7.4 TWh), and the total "working" capacity in all the structures combined was estimated to be 228.2 Mt or 89.8 TWh. The $H_2$ cushion gas was estimated for 13 UGS facilities to be in the range of 1.5–122.6 Mt (average 17.8 Mt), with a total of 231.27 Mt.

The $H_2$ and $CO_2$ storage capacities estimated here represent the first step in calculating the theoretical capacity that could be stored. It will not be possible to use the full volume since factors such as injectivity, irreducible water saturation, unrecoverable gas, and permeability variations will affect the amount of storage capacity that can be accessed at useful rates and within economic constraints (e.g., technically accessible $CO_2$ storage resource calculations in $CO_2$StoP for $CO_2$ storage) [44].

The accurate estimation of the $H_2$ storage capacity of a geological structure is a complex process that requires a multidisciplinary approach, specialised tools, and detailed site data to ensure safe and efficient operations. The presence of cushion gas in existing UGS facilities may affect the quality of the stored $H_2$.

$H_2$ has different physical and chemical properties than natural gas, and the cushion gas may mix with the stored $H_2$. This could lead to operational issues. Additionally, $H_2$ molecules can permeate materials more easily than natural gas and $CO_2$, which means that the integrity of the seal must be evaluated to confirm its effectiveness as a caprock for $H_2$ [44]. More detailed reservoir simulations and the pilot testing of this concept are required.

The issue of cushion gas in $H_2$ storage at UGS facilities necessitates a thorough approach that encompasses site integrity, compatibility, efficiency, and regulatory compliance to effectively transition from natural gas to $H_2$ storage.

## 5. Discussion

This study shows that the existing UGS facilities in Ukraine could play a crucial role in $H_2$ and $CO_2$ storage not only for Ukraine but also for Europe. Ukraine has a well-developed network of UGS facilities (13), with a working gas capacity of about 32 BCM, located in the oil–gas-rich western, central, eastern, and southern regions.

As part of the Hystories project, a detailed evaluation of the geology and petrophysical properties of UGS facilities was carried out, and their $H_2$ storage capacity was estimated in energy units. The Hystories database contains all the geological and petrophysical characteristics of the reservoirs and seals in the investigated UGS reservoirs, including thier age, formation, temperature, pressure, porosity, permeability, thickness, lithology, and depth. In this study, all the necessary parameters for calculating $H_2$ and $CO_2$ storage capacities were updated with the latest data, and the storage capacities for $H_2$ and $CO_2$ were estimated.

The total estimated $H_2$ storage capacity in all the studied UGS storage sites in energy units (maximum probability assessment) is about 98.8 TWh and 459.6 or around 228.2 Mt, considering the replacement of the total and working gas volumes, respectively. The total estimated $CO_2$ storage capacity in all the structures is approximately 37.6 Gt for optimistic scenarios and 18.8 Gt for conservative scenarios.

The significant capacities of existing onshore UGS reservoirs built on depleted gas/gas condensate fields and saline aquifers indicate that the UGS system could be the main potential location for $H_2$ and $CO_2$ storage in Ukraine and could make Ukraine a major player in the $H_2$ and $CO_2$ storage market. The depleted hydrocarbon traps are found in Palaeozoic (Carboniferous and Permian), Mesozoic (Triassic, Jurassic, and Cretaceous), and Cenozoic (Paleogene and Neogene) strata.

The UGS reservoirs, which comprise sandstones, limestones, and dolomites, have good reservoir properties: a porosity of 7 to 31% and a permeability of 8 up to 24 mD. The UGS seals consist of claystone, salt, siltstones, and anhydrites.

The geological features of the strata in Ukraine are promising for $H_2$ and $CO_2$ storage. These potential reservoirs have net thicknesses that range from 8 to 187 m and depths that

range from 580 to 1210 m. It can be concluded that the existing UGS facilities in Ukraine are promising options for storing $H_2$ and $CO_2$.

Based on recently published techno-economic modelling of $H_2$ storage in European UGS facilities, several Ukrainian UGS facilities (Chervonopartyzanske, Dashavske, Oparske, and Bilche–Volytsko–Uherske) are considered the most favourable for $H_2$ storage in Europe [14]. However, it is also mentioned that UGS facilities could be repurposed in the future not only for $H_2$ or $CO_2$ storage but also for biomethane ($CH_4$) storage. There are several recent studies considering the use of $CO_2$ as a cushion gas and comparing this with $CH_4$ and Nitrogen ($N_2$) with different and sometimes contradictory results [45]. In addition to these studies, different gas mixtures have been modelled, with the authors concluding that the ideal cushion gas for $H_2$ storage is a mixture of $H_2$ (50%), $CO_2$ (40%), $CH_4$ (5%), and $N_2$ (5%) [46]. It was also reported [47] that the use of a high proportion of $CO_2$ in the cushion gas of sandstone reservoirs could minimise the risks associated with $H_2$ storage projects.

In our study, the amount of $CO_2$ that would replace the cushion gas for $H_2$ storage was calculated separately and, therefore, could be applied to the $H_2$ storage scenario we present, replacing the $H_2$ cushion gas with $CO_2$ cushion gas. The minimum depth required for $H_2$ storage is reported as 305 m [48,49]. The minimum depth required for $CO_2$ geological storage is usually reported as 800 m for dense phase $CO_2$ storage (>31.1 °C and >7.38 MPa) [50]. However, $CO_2$ could also potentially be stored in a liquid state at a lower temperature and at a shallower depth, although it would have a higher density and viscosity. In several of the investigated Ukrainian UGS reservoirs with a temperature range of 23–27 °C and a depth in the range of 660–770 m, the $CO_2$ could be stored in a liquid state (Table 4) with high-density efficiency for $CO_2$ storage. Considering the different depth requirements for $CO_2$ and $H_2$ storage, the shallower UGS facilities, such as Olyshivske, Chervonopartyzanske, and Krasnopopivske, are less favourable for $CO_2$ storage but could be suitable for $H_2$ or $CH_4$ storage.

## 6. Conclusions

The UGS facilities in Ukraine have the potential to serve as valuable assets for both $H_2$ and $CO_2$ storage (including using $CO_2$ as a cushion gas), contributing to various aspects of the energy transition and sustainability efforts. The repurposing of UGS facilities for $H_2$ and $CO_2$ storage requires careful planning, safety measures, and compliance with legal regulations. Additionally, fluid compatibility assessments and seal assessments are essential to ensure the safe containment of these fluids.

Underground $H_2$, biomethane, and $CO_2$ storage play important roles in transitioning to a low-carbon economy, reducing greenhouse gas emissions, and ensuring affordable, clean, and modern energy while enhancing energy security. Increasing the share of renewable energy and integrating sustainable development across various sectors of the economy is crucial for achieving climate goals. $H_2$ production will significantly promote the environmental, climate, and social dimensions of sustainable development by reducing $CO_2$ emissions, enhancing energy security, and creating new job opportunities.

Additionally, considering Ukraine's location and extensive experience in UGS, Ukraine could play a key role in developing a pan-European hydrogen economy.

**Author Contributions:** Conceptualization, Y.D., A.S. and B.M.; methodology, A.S., K.S. and C.J.V.; software, K.S.; validation, A.S., B.M., K.S. and C.J.V.; formal analysis, Y.D., A.S., K.S. and B.M.; investigation, Y.D.; resources, Y.D.; data curation, Y.D.; writing—original draft preparation, Y.D., A.S. and B.M.; writing—review and editing, Y.D., A.S., B.M., K.S. and C.J.V. All authors have read and agreed to the published version of the manuscript.

**Funding:** This work was supported by the Hystories project (https://hystories.eu/, accessed on 2 March 2025) and funded by the Fuel Cells and Hydrogen 2 Joint Undertaking (now Clean Hydrogen Partnership) under grant agreement No 101007176. This Joint Undertaking receives support from the European Union's Horizon 2020 research and innovation programme and Hydrogen Europe and Hydrogen Europe Research.

**Institutional Review Board Statement:** Not applicable.

**Informed Consent Statement:** Not applicable.

**Data Availability Statement:** For existing underground gas storage facilities, the available geological and petrophysical data were collected from ESTMAP Project–Energy Storage Mapping and Planning and https://geoinf.kiev.ua/wp/index.html (accessed on 2 March 2025). Ceri Vincent publishes with the permission of the Executive Director of the British Geological Survey (BGS, UKRI).

**Conflicts of Interest:** Author Yuliia Demchuk is not employed by NGO Geothermal Ukraine but serves as a Board member. Author Kazbulat Shogenov was employed by the Tallinn University of Technology and SHOGenergy Consulting. Author Alla Shogenova was employed by the Tallinn University of Technology and SHOGenergy Consulting. Author Barbara Merson was employed by the National Institute of Oceanography and Applied Geophysics. Author Ceri Jayne Vincent is an employee of United Kingdom Research and Innovation as represented by British Geological Survey. The remaining authors declare that the research was conducted in the absence of any commercial or financial relationships that could be construed as a potential conflict of interest.

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
