# Peer review of "Geological and Petrophysical Properties of Underground Gas Storage Facilities in Ukraine and Their Potential for Hydrogen and CO2 Storage"

_sustainability, doi:10.3390/su17062400_

Round 1
Reviewer 1 Report (Previous Reviewer 1)
Comments and Suggestions for Authors
This manuscript has greatly improved since the first time submission. Suggestions include : 1. Large blank space between paragraphs should be removed.
2. One or two sentences should not be a paragraph without a distinct major idea and should be combined wtih other paragraphs with closed meaning. For example:(1) In line 520, "Latitudinal and meridional faults determined the block structure of the crystalline basement." I cannot understand what does the author want to express by putting a separate sentence here. (2) "To calculate H2 storage capacity in energy units the heating value for H2 of 39.4 724 kWh/kg was applied [41]." This sentence should be moved to previous paragraphs stating the calculation of H2 storage capacity.
3. Accuracy should be improved. One example is : VH2 active = Vactive ∗ρH2r, on the left of this equation, V should be changed to M
The quality of english is fine. Some comments please refer to the above.
Author Response
Comment 1: 1. Large blank space between paragraphs should be removed
Reply: Done!
Comment 2: One or two sentences should not be a paragraph without a distinct major idea and should be combined wtih other paragraphs with closed meaning. For example:(1) In line 520, "Latitudinal and meridional faults determined the block structure of the crystalline basement." I cannot understand what does the author want to express by putting a separate sentence here. (2) "To calculate H2 storage capacity in energy units the heating value for H2 of 39.4 724 kWh/kg was applied [41]." This sentence should be moved to previous paragraphs stating the calculation of H2 storage capacity.
Reply: Done!
Comment 3: Accuracy should be improved. One example is : VH2 active = Vactive ∗ρH2r, on the left of this equation, V should be changed to M
Reply: Done!
Reviewer 2 Report (New Reviewer)
Comments and Suggestions for Authors
Manuscript ID: sustainability-3324560
The manuscript entitled “Geology and Petrophysical Properties of UGSs in Ukraine and Their Potential for Hydrogen and CO2 Storage” is well written, organized, and referenced. The overall structure of the paper is relatively reasonable, the problem and aim of the study are clearly obvious and related to the scope of the sustainability Journal and can be accepted for publication after making the following minor corrections. Benefits
1- The resolution of some Figures appears to be of poor quality, Figures resolutions should be enhanced (should be in 300 – 600 dpi).
2- The Keywords is too much, I think it should be reduced to 6 keywords enough.
3- Move Figure 1 from Introduction section to Geological background section.
4- The Figure caption of most figures is too large, for example, the caption of Figure 2 contains an explanation of the legend, so there is no need to explain it in the caption as you defined it in the Figure.
5- Figures 3 & 5 are the same as above.
6- In Figure 6 insert a legend to the figure and remove the definitions of the lagend form the figure caption.
7- The same as above for Figures 7, 10, 12 and 13.
8- It is better to state the advantages and limitations of your proposed methodology in the conclusion section.
9- The rest is OK.
Best wishes,
Author Response
Comment 1: The resolution of some Figures appears to be of poor quality, Figures resolutions should be enhanced (should be in 300 – 600 dpi).
Reply: Done as much as possible.
Comment 2: The Keywords is too much, I think it should be reduced to 6 keywords enough.
Reply: Done! Reduced to 6.
Comment 3: Move Figure 1 from Introduction section to Geological background section.
Reply: Done!
Comment 4: The Figure caption of most figures is too large, for example, the caption of Figure 2 contains an explanation of the legend, so there is no need to explain it in the caption as you defined it in the Figure.
Reply: Done! Removed
Comment 5: Figures 3 & 5 are the same as above.
Reply: Done! In Figure 3 caption text was deleted, in Figure 5 we added abbreviations to the figures' legend.
Comment 6: In Figure 6 insert a legend to the figure and remove the definitions of the lagend form the figure caption.
Reply: Done!
Comment 7: The same as above for Figures 7, 10, 12, and 13.
Reply: Done! Figures 7, 10, and 12 have been updated with legends. Figure 13 was not updated due to excessive text in the caption.
Reviewer 3 Report (New Reviewer)
Comments and Suggestions for Authors
Dear Authors,
Thank you. It is a valuable and important study.
The work is well organised. The MS is within the scope of the journal, scientifically valid; mathematically, technically and experimentally accurate, and the tackled problem is of global importance. In general, the study will give valuable information about the potentiality of Underground storage of H2 and CO2 in Ukraine and Europe. The work’s global importance is high.
Comments
I have some minor comments.
1- Line 40, Please discuss on a domestic level, factors influencing GHGE.
2- Line 57, Please give international examples (more literature) of the decarbonization of the energy sectors. and discuss its present methodologies in these countries.
3- Tables 2,3,&4, An in-depth statistical model for this large amount of data could give new results and conclusions.
Regards
Author Response
Comments 1 and 2:
1- Line 40, Please discuss on a domestic level, factors influencing GHGE.
2- Line 57, Please give international examples (more literature) of the decarbonization of the energy sectors. and discuss its present methodologies in these countries.
Reply to 1 and 2:
We acknowledge the reviewer's feedback and agree that including these details would enhance the paper. However, earlier reviewers had requested that we omit this information, as they felt it made the text too dense. Therefore, this comment contradicts the feedback we received from the previous reviewers.
Comments 3: Tables 2,3,&4, An in-depth statistical model for this large amount of data could give new results and conclusions.
Reply 3: Agreed with this comment. Tables 2-4 have been updated with statistical analysis, and additional text has been added to paragraph 4.
This manuscript is a resubmission of an earlier submission. The following is a list of the peer review reports and author responses from that submission.
Round 1
Reviewer 1 Report
Comments and Suggestions for Authors
This manuscript is meaningful by evluating the potential of Hydrogen and CO2 Storage based on the geology and petrophysicalproperties of underground gas storage facilities in Ukraine. As a whole, this manuscript summarizes the data and method, Geological background and location of UGS in Ukraine and Potential for CO2 and H2 storage based on 45 references, which is good. However, there are two main problems. The first is the writing is not concise enough. Second the security of Hydrogen and CO2 Storage is not discussed at depth based on technical view.
Comments on the Quality of English Language
Examples for being not concise:
1. In the abstract, the words on significance of Underground storage of H2 andCO2 are too much, which is unnecessary. From Line 24 "important roles" until 29 ' and creating new job opportunities." More than 6 lines.
2. There are 10 keywords, which are too much. The keywords such as "energy transition; energy security; clean energy" can be removed because those are not mainly discussed in the manuscript.
Author Response
Comment 1: In the abstract, the words on significance of Underground storage of H2 andCO2 are too much, which is unnecessary. From Line 24 "important roles" until 29 ' and creating new job opportunities." More than 6 lines.
Response: Accepted, some sentences were shortened and removed
Comment 2: There are 10 keywords, which are too much. The keywords such as "energy transition; energy security; clean energy" can be removed because those are not mainly discussed in the manuscript.
Response: done!
Reviewer 2 Report
Comments and Suggestions for Authors
1. There are too many keywords, usually 5 to 8.
2. Line 106. The citation order is incorrect.
3. The introduction is not well written, does not conform to the norms of academic paper writing. It has too many research backgrounds, and there are few domestic and foreign research status related to underground gas storage. Few references are cited. The research background should be simplified and the progress of evaluation, site selection and renovation of underground gas storage closely related to the research content should be supplemented.
4. According to the conventional thesis writing logic, the geological background should be in Chapter 2, and the data and methods in Chapter 3.
5. In Fig. 5. The full name of the abbreviation should be marked where it first appears.
6. In Fig. 6. The necessary geological map elements such as legend, north pointer and scale are lacking.
7. The 4 chapter introduces the stratigraphy and lithology of the eastern and western regions without mentioning the petrophysical properties such as porosity, permeability and rock mechanical strength.
8. The discussion and the conclusions should be separated. The discussion is the most important part in the paper and it represent the academic level of your study.
9. Figure 14 should be placed where these concepts appear first.
Comments on the Quality of English Language
I have no comments on the quality of language.
Author Response
Comment 1: There are too many keywords, usually 5 to 8.
Response: Done, removed
Comment 2: Line 106. The citation order is incorrect.
Response: Improved
Comment 3: The introduction is not well written, does not conform to the norms of academic paper writing. It has too many research backgrounds, and there are few domestic and foreign research status related to underground gas storage. Few references are cited. The research background should be simplified and the progress of evaluation, site selection and renovation of underground gas storage closely related to the research content should be supplemented.
Response: Іntroduction has been shortened, rephrased, and supplemented with current projects/research.
Comment 4: According to the conventional thesis writing logic, the geological background should be in Chapter 2, and the data and methods in Chapter 3.
Response: Implemented
Comment 5: In Fig. 5. The full name of the abbreviation should be marked where it first appears.
Response: Done
Comment 6: In Fig. 6. The necessary geological map elements such as legend, north pointer and scale are lacking.
Response: Done, in all figures the legend and north pointer were added
Comment 7: The 4 chapter introduces the stratigraphy and lithology of the eastern and western regions without mentioning the petrophysical properties such as porosity, permeability and rock mechanical strength.
Response: This data are described in table 2
Comment 8: The discussion and the conclusions should be separated. The discussion is the most important part in the paper and it represent the academic level of your study.
Response: Done
Comment 9: Figure 14 should be placed where these concepts appear first.
Response: Done, replaced to the 1.Introduction
Reviewer 3 Report
Comments and Suggestions for Authors
While the subject matter is significant in the context of transitioning to low-carbon energy systems, I am compelled to reject the manuscript for the following reasons:
The data presented on the analyzed underground gas storage facilities is primarily sourced from older references, with the majority dating back to the late 1990s and early 2000s. Only three references (8, 11, and 13) from 2011 are relatively recent, but I was unable to access them online. It is essential for the paper to include accessible, up-to-date references to ensure the validity and relevance of the data used. The reliance on older datasets raises concerns about whether the findings are representative of the current state of these UGS facilities.
In the data and methods section, there is explanation of clear parameters , such as the unrecoverable gas in UGS facilities. Additionally, the paper only mentions volume multiplied by density in equations 1 to 4, without any clarification on other factors that may influence storage capacity.
The discussion on UGS facilities in Ukraine is mentioned both in the Introduction and the Geological Background. These sections could be merged to avoid redundancy and streamline the paper’s flow.
The paper mainly compiles and reviews data on six oil and gas-bearing stratigraphic units without introducing significant innovation and finally suggest that the Bilche−Volytsko−Uherske storage site has the best storage capacity based on size, porosity, and permeability. There is no detailed explanation of the depletion status of the site or new insights into optimizing the UGS for hydrogen and CO2 storage beyond basic criteria.
Comments on the Quality of English LanguageMinor editing of English language required.
Author Response
Comment 1: The data presented on the analyzed underground gas storage facilities is primarily sourced from older references, with the majority dating back to the late 1990s and early 2000s. Only three references (8, 11, and 13) from 2011 are relatively recent, but I was unable to access them online. It is essential for the paper to include accessible, up-to-date references to ensure the validity and relevance of the data used. The reliance on older datasets raises concerns about whether the findings are representative of the current state of these UGS facilities.
Response:
Due to the geopolitical situation and Martial Law in Ukraine since February 24, 2022, and in accordance with Subsection 4, Paragraph 1 of the Cabinet of Ministers of Ukraine Resolution No. 263, dated March 12, 2022, titled "Some Issues of Ensuring the Functioning of Information and Communication Systems, Public Electronic Registers under Martial Law," access to public registers and online databases has been restricted to safeguard national security.
Reports and data on underground gas storage facilities from the late 1990s and early 2000s can be obtained by prior request and accessed by visiting the reading room of the State Research and Development Enterprise “Geoinform of Ukraine”. However, the new reports from 2011 are proprietary of the UGS operator, are commercial, and are not publicly available.
Comment 2: In the data and methods section, there is explanation of clear parameters , such as the unrecoverable gas in UGS facilities. Additionally, the paper only mentions volume multiplied by density in equations 1 to 4, without any clarification on other factors that may influence storage capacity.
Response:
We are mentioning that: "H2 density in-situ reservoir conditions (ρH2r) was calculated as a function of pressure and temperature using models for thermodynamic properties of pure fluids [40]. “, underlining that pressure and temperature are the main factors of any gases or liquids injected and stored underground, influencing storage capacity. In addition, we implemented minimum and maximum pressure values to model different reservoir conditions where H2 or CO2 will be stored.Comment 3: The discussion on UGS facilities in Ukraine is mentioned both in the Introduction and the Geological Background. These sections could be merged to avoid redundancy and streamline the paper’s flow.’
Response: The introduction was reviewed and changed
Comment 4: The paper mainly compiles and reviews data on six oil and gas-bearing stratigraphic units without introducing significant innovation and finally suggest that the Bilche−Volytsko−Uherske storage site has the best storage capacity based on size, porosity, and permeability. There is no detailed explanation of the depletion status of the site or new insights into optimizing the UGS for hydrogen and CO2 storage beyond basic criteria.
Response: Detailed explanation on status, new insights, and UGS operator plans were added.